# acuSimNet: Multi-View Self-Occlusion-Aware Visibility Learning for Cranio-Cervical Acupuncture Points

## Abstract

The localization of acupuncture points (acupoints) in Traditional Chinese Medicine (TCM) presents unique challenges since they are defined by abstract principles rather than distinct anatomical landmarks. Existing approaches are typically constrained to single view or rely on indirect calculations of relative positions with respect to other landmarks, thereby overlooking human anatomical variations. Furthermore, acupoint visibility assessment, which determins whether points are occluded by human itself, hair, clothing, or other objects, has received limited attention in practical applications. Acupoint localization itself does not require 3D reconstruction, but inferring their occlusion relationships with anatomical surfaces does, which adds computational cost and limits real-time inference. In this work, we introduce *acuSimNet*, an efficient hierarchical multi-task learning architecture for multi-view, self-occlusion-aware visibility prediction of acupoints, achieving 99.97% accuracy on the validation set. Our approach also addresses the challenges of high-dimensional classification (174 acupoints in cervicocranial region), negative convergence issues for visible acupoints, and intertask scheduling optimization, resulting in substantially accelerated convergence. We improved the training efficiency from the exisiting methods of 3000 epochs to achieve 99% validation accuracy, to our optimized framework achieving 90% accuracy in 39 epochs and 99% accuracy in 86 epochs. This architecture overcomes the limitations of existing methods, could enable practical applications in acupoints detection and visualization, advancing the automation of TCM.

## 1 Introduction

Traditional Chinese Medicine (TCM) acupuncture represents an established therapeutic modality whose clinical efficacy fundamentally depends on precise identification and stimulation of specific anatomical landmarks known as acupuncture points or acupoints Wang et al. (2018). Each acupoint is associated with distinct therapeutic effects, requiring accurate stimulation to facilitate the arrival of *qi* (energy flow) and remove physiological blockages of Traditional Chinese Medicine (Beijing); Lim et al. (2010). In accordance with TCM principles, acupoints and meridian distributions exhibit substantial variation among individuals due to anatomical differences including height, weight, gender, and other morphological factors Deadman et al. (2007); for the Western Pacific (2008). The safety considerations inherent in acupuncture practice necessitate exceptional precision in acupoint localization, as incorrect needle placement due to occluded or misidentified points can lead to serious clinical complications, distinguishing this task from conventional computer vision applications where prediction errors typically result in reduced accuracy rather than potential patient harm.

However, existing acupoint localization approaches suffer from significant limitations including single-viewpoint constraints, low-dimensional classification tasks, and complete neglect of visibility assessment Zhang et al. (2023); Sun et al. (2020); Masood & Qi (2022); Ji & Zhou. Recent advances in domain-specific synthetic data generation for acupuncture, particularly the *acuSim* dataset Sun et al. (2025) provides comprehensive multi-view visibility annotations for 174 cranio-cervical acupoints across 63,936 images that serves as the benchmark for our optimization work. However, existing neural network approaches applied to this dataset lack explicit visibility learning objectives, resulting in training instability and suboptimal convergence characteristics.

Beyond domain-specific limitations, the broader field of human landmark detection has similarly exhibited limited attention to visibility assessment and occlusion handling. Traditional approaches, including regression-based methods like DeepPose Toshev & Szegedy (2014) and early CNN-based

pose estimation Jain et al. (2014), directly predict landmark coordinates through neural network regression. Conversely, heatmap-based methods, demonstrated in works like the Deep Alignment Network Kowalski et al. (2017) and boundary-aware approaches Wu et al. (2018), generate probability distributions over spatial locations but similarly lack explicit visibility modeling. Methods such as VNect Mehta et al. (2017) for real-time 3D pose estimation and integral regression approaches Sun et al. (2018) have advanced coordinate prediction accuracy but remain constrained by the assumption of perpetual landmark visibility. Additionally, unsupervised landmark discovery methods Zhang et al. (2018) have emerged to automatically identify structural representations, but these approaches also do not explicitly model landmark visibility or occlusion states.

Medical imaging applications have explored anatomically-structured landmark detection through spatial configuration modeling Payer et al. (2019) and universal multi-domain frameworks Lu et al. (2024), yet these methods target skeletal landmarks in X-ray imaging rather than surface-based anatomical points subject to soft-tissue occlusion. Multi-person pose estimation systems employing bottom-up approaches Papandreou et al. (2018) and temporal tracking Xiu et al. (2018) have demonstrated sophisticated keypoint association mechanisms, but their confidence scores represent detection uncertainty rather than true semantic visibility of occluded points. The multi-task cascaded convolutional networks approach, as implemented in MTCNN Zhang et al. (2016), represents an advancement in joint face detection and landmark localization but remains limited to a small set of five facial landmarks and provides no visibility assessment capabilities. More recent developments, including transformer-based approaches Li et al. (2022), direct regression methods Mao et al. (2022), lightweight detectors Guo et al. (2019), and hybrid architectures Jin et al. (2021), while achieving impressive accuracy on benchmark datasets, continue to operate under the assumption of complete landmark visibility, failing to address practical challenges posed by occlusion or extreme poses.

Our study proposes ***acuSimNet***, a hierarchical multi-task framework that addresses key neural network design constraints in current localization approaches such as *acuSim* Sun et al. (2025). Our approach explicitly models visibility prediction, coordinate regression, and acupoint classification within a unified framework that incorporates TCM domain knowledge through meridian-specific processing branches. This work represents a domain-driven system-level engineering effort rather than fundamental algorithmic innovation. Our contributions are as follows: first, we address neural network design limitations in existing acuSim applications through explicit visibility learning objectives, achieving 35× convergence acceleration (from 3000 to 86 epochs for 99% accuracy); second, we engineer a hierarchical meridian-specific architecture that reduces classification complexity by incorporating medical domain knowledge, enabling robust multi-view detection; and third, we systematically integrate established techniques including dynamic soft visibility masking and uncertainty-based task weighting Kendall et al. (2018), optimizing their coordination for acupuncture-specific challenges to achieve clinically relevant 0% false positive rates.

## 2 RELATED WORK

Current acupoint localization research demonstrates significant limitations that restrict clinical applicability. Existing acupoint datasets Zhang et al. (2023); Sun et al. (2020); Masood & Qi (2022); Ji & Zhou are characterized by small-scale implementations with severely limited annotations. For example, Ji & Zhou annotated only four acupoints on the upper back, while other studies Masood & Qi (2022) constructed datasets with merely 600 RGB-D images focusing exclusively on hand acupoints. These approaches suffer from three fundamental limitations: first, they address only low-dimensional classification problems with fewer than 30 acupoints, significantly below the 174 points present in the cranio-cervical region alone; second, they are constrained to fixed viewpoints or frontal images, failing to account for multi-view scenarios encountered in clinical practice; and third, they completely neglect acupoint visibility assessment, assuming all targeted acupoints are perpetually visible and accessible for treatment.

The computational methods employed in existing acupuncture localization systems Chang & Zhu (2017); Jiang et al. (2016) typically follow a two-stage approach: initial detection of facial features followed by calculation of acupoint locations based on predetermined anatomical relationships. However, these methods are fundamentally limited to frontal face images and lack the angular awareness necessary for robust performance across varying poses and viewpoints. Similarly, approaches targeting anatomically simpler structures like forearms Simonyan & Zisserman (2014); Chan et al. (2021) demonstrate inadequate consideration of the complex three-dimensional relationships and

occlusion patterns characteristic of cranio-cervical anatomy. As shown in Table 3, existing acupoint datasets demonstrate significant scale limitations compared to comprehensive approaches.

Existing frameworks for landmark visibility assessment, most notably MediaPipe Lugaresi et al. (2019), provide insights into multi-view landmark detection challenges. MediaPipe outputs confidence scores for each detected landmark, representing detection reliability rather than true semantic visibility. While these scores serve as visibility proxies, they fundamentally represent prediction uncertainty rather than occlusion-aware assessment. The inherent geometric properties of landmark detection create fundamental challenges for occlusion modeling. Unlike objects with physical volume, landmarks represent discrete spatial locations that can only be occluded by higher-order anatomical structures like facial surfaces, body segments, or external objects like clothing and hair.

The sparse distribution of standard anatomical landmarks detected by general-purpose systems creates complications for acupuncture applications. While MediaPipe provides visibility confidence for standard facial and body landmarks, the vast majority of acupoints do not coincide with these conventional landmarks. The substantial spatial gaps between standard landmarks and acupoints locations mean that visibility assessments cannot be reliably extrapolated. This limitation is particularly problematic given the safety-critical nature of acupuncture practice, where incorrect assumptions about point visibility could lead to inappropriate needle placement and potential patient harm.

Recent advances Sun et al. (2025) in synthetic data generation have opened new possibilities for comprehensive acupuncture acupoint detection, addressing data scarcity and annotation cost challenges that have historically limited progress in this domain. Synthetic data generation enables the creation of diverse anatomical variations, multiple viewing angles, and controlled occlusion scenarios that would be prohibitively expensive to collect from human subjects. However, existing approaches treat visibility labels merely as supervisory signals for filtering training samples rather than incorporating visibility prediction as an explicit learning objective within the network architecture. This design limitation prevents models from developing independent visibility assessment capabilities, instead relying on ground truth visibility annotations as prior during training.

The absence of explicit visibility prediction mechanisms results in suboptimal learning dynamics, where models struggle to distinguish between coordinate prediction tasks for visible versus occluded acupoints. These approaches exhibit severely impaired convergence characteristics, often requiring extensive training periods to achieve acceptable performance levels. The failure to explicitly model visibility as a learnable task represents a critical gap that limits practical applicability in real-world clinical scenarios where ground truth visibility information is unavailable.

## 3 METHODOLOGY

### 3.1 ARCHITECTURE OVERVIEW

In this section, we proceed to explain the overall architecture of the proposed acuSimNet, which consists of a backbone convolutional neural network followed by nine parallel meridian-specific processing branches, as illustrated in Figure 1. Among the fourteen major meridians defined in TCM, nine traverse the cervicocranial region, and each of these is represented by one processing branch. The backbone network, implemented using DenseNet201 Huang et al. (2017), processes input images through systematic dimensionality transformation: from input tensor $(1, 3, 512, 512)$ through convolutional feature extraction to $(1, 1920, 16, 16)$, followed by global average pooling and flattening to a 1920-dimensional global feature vector.

The architecture design philosophy centers on adapting standard neural network structures to TCM domain knowledge: acupoints exhibit hierarchical relationships within meridian systems, which can be leveraged to reduce classification complexity. Unlike conventional approaches that treat all acupoints equally, our engineering approach recognizes that acupoints within the same meridian share anatomical and functional relationships that guide the learning process.

### 3.2 HIERARCHICAL MULTI-TASK LEARNING FRAMEWORK

The hierarchical multi-task learning framework represents a core engineering design of acuSimNet, designed to address three fundamental challenges in acupuncture acupoint detection: visibility assessment, precise localization, and accurate classification. Rather than treating these as independent tasks, our framework models their interdependencies through a carefully designed hierarchy that reflects clinical decision-making in TCM practice, applying established multi-task learning principles to domain-specific requirements.

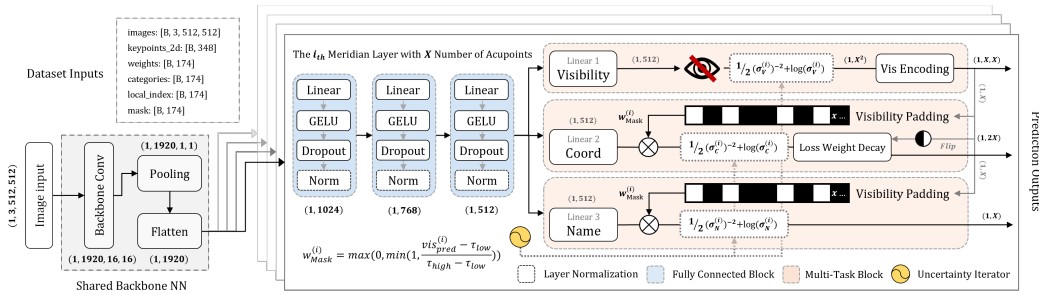

Figure 1: The Overall Architecture of acuSimNet

The framework operates on the principle that visibility determination should precede and inform both localization and classification decisions, mirroring the clinical reasoning process where practitioners first assess point accessibility before determining precise location and identity. The framework processes RGB images with tensor shape $[B, 3, 512, 512]$ alongside ground truth annotations including 2D coordinates $[B, 348]$ representing 174 acupoints, visibility masks $[B, 174]$ indicating occlusion states, acupoint categories $[B, 174]$ for meridian classification, and local indices $[B, 174]$ for intra-meridian positioning.

Each meridian branch processes the global feature representation through a shared fully connected layer that maps 1920-dimensional backbone features to a 512-dimensional meridian-specific representation. This serves as the foundation for three specialized prediction heads: visibility classification, coordinate regression, and acupoint classification. The hierarchical nature becomes evident in the information flow between tasks, where visibility prediction head confidence scores modulate loss calculations for coordinate regression and classification tasks through dynamic weighting mechanisms, ensuring the model focuses computational resources on acupoints likely to be visible and clinically relevant.

### 3.3 Medical Prior-based Meridian Layering

The meridian layering strategy represents a domain-specific architectural adaptation that incorporates TCM domain knowledge into standard neural network structures, addressing the limitation of conventional acupoint detection methods that ignore anatomical and functional relationships between acupoints. This engineering approach recognizes that acupoints follow systematic patterns defined by meridian theory, refined over thousands of years of clinical practice.

Each meridian layer is designed to handle specific characteristics of its corresponding meridian system. The layer architecture adapts to varying numbers of acupoints per meridian, with output dimensions configured to match the maximum number of points in each system. For example, the Gallbladder (GB) meridian layer accommodates up to 42 acupoints in the cervicocranial area, while the Ren (RN) meridian handles only 2 points. This adaptive sizing prevents model domination by meridians with larger point numbers while ensuring smaller meridian systems receive adequate representational capacity.

The medical prior integration extends beyond structural organization to influence learning dynamics. Each meridian layer maintains independent learnable parameters for task weighting, allowing the model to discover meridian-specific optimal balances between visibility prediction, coordinate regression, and classification tasks. This design acknowledges that different meridian systems may require different task emphasis based on their anatomical locations and clinical accessibility patterns. The layering strategy also facilitates clinical interpretation by organizing predictions according to medical categories, enabling direct mapping to clinical terminology and treatment protocols.

### 3.4 Dynamic Visibility Learning Mechanism

The dynamic visibility learning mechanism addresses one of the most challenging aspects of acupuncture acupoint detection: determining whether specific points are visible and accessible in a given view. Unlike conventional human acupoint detection tasks, visibility assessment represents the most critical factor in acupoints detection due to safety considerations and the interactive nature of acupuncture treatment. In clinical practice, false positive cases (Type I Error[1] Neyman & Pearson

---

[1] In statistical hypothesis testing, the error of rejecting a null hypothesis when it is actually true.

(1992), where occluded points are incorrectly predicted as visible, potentially leading to misplaced needle insertions) pose significantly greater risks than false negative cases (Type II Error[2], where visible points are predicted as occluded, resulting only in missed treatment opportunities). Acupoints can be occluded by hair, clothing, body positioning, or anatomical variations, requiring sophisticated visibility assessment capabilities.

### 3.4.1 VISIBILITY PREDICTION HEAD

The visibility prediction head serves as the primary visibility classification strategy, implementing explicit binary classification to distinguish between visible and occluded acupoints. This head processes 512-dimensional meridian-specific features through a dedicated fully connected layer that outputs visibility probabilities for each potential acupoint location within the meridian, producing a tensor of shape $(1, X)$ where $X$ represents the maximum number of acupoints in the meridian.

The visibility prediction operates independently of coordinate and classification tasks, ensuring unbiased visibility assessment. The head produces probability scores using sigmoid activation, yielding values between 0 and 1 representing the model's confidence in acupoint visibility. These scores serve dual purposes: providing direct visibility predictions for clinical applications and generating dynamic weights for coordinating other task learning.

The design emphasizes robustness to various occlusion scenarios encountered in clinical practice. The head must distinguish between self-occlusion (where body postures or viewing angles cause points to be obscured), external occlusion (clothing, hair coverage, medical equipment), and inherent invisibility (points located on non-visible body surfaces like lingual acupoints). Hair occlusion represents a particularly common challenge, making it critical for clinical applications.

### 3.4.2 COORDINATE REGRESSION HEAD

The coordinate regression head implements implicit visibility learning while focusing on precise acupoint localization for visible points. This dual functionality addresses extreme data imbalance scenarios where nearly all acupoints in certain views may be occluded. For example, when observing a patient with dense hair coverage from an overhead perspective, only a few acupoints on exposed skin areas remain visible, while the majority of cranial points are completely obscured.

The head employs an implicit clustering strategy that converges occluded acupoints to the coordinate origin (0,0) while accurately predicting spatial coordinates for visible acupoints within the facial region. The coordinate regression integrates visibility information through a sophisticated weighting mechanism: the visibility encoding from the visibility head undergoes visibility padding, which is then element-wise multiplied with coordinate predictions. Additionally, a loss weight decay mechanism specifically targets false positive cases (invisible points incorrectly predicted as visible) through a specialized weighting function, producing the final coordinate output of shape $(1, 2X)$ representing $(x, y)$ coordinates for $X$ acupoints.

This approach provides a natural complementary mechanism to explicit visibility predictions. The clustering-like behavior emerges naturally as the model learns to minimize localization errors for visible points while finding consistent representation for false positive cases. The coordinate regression utilizes normalized coordinates in the range [0,1] to ensure scale invariance across different image sizes and camera configurations, facilitating generalization across various imaging setups commonly encountered in clinical environments.

### 3.4.3 CLASSIFICATION HEAD

The classification head performs meridian-specific acupoint identification, taking advantage of reduced classification complexity achieved through meridian layering. Instead of classifying among all 174 possible acupoints simultaneously, each classification head operates within its meridian-specific subset, significantly reducing classification complexity and improving learning efficiency. The classification process incorporates visibility information through the same visibility padding mechanism, where visibility encoding modulates classification predictions through element-wise multiplication, producing the final output of shape $(1, X)$ representing class probabilities for $X$ meridian-specific acupoints.

---

[2]In statistical hypothesis testing, the error of failing to reject a null hypothesis when it is actually false.

The head architecture adapts to varying numbers of acupoints within each meridian, with output dimensions matching the maximum number of points in the corresponding meridian system. This adaptive design ensures that softmax probability distributions are properly normalized within each meridian context, preventing bias toward meridians with larger numbers of acupoints. The classification process incorporates hierarchical medical knowledge embedded in meridian theory, as points within the same meridian often share anatomical relationships and functional properties that the model can leverage to improve classification accuracy.

## 3.5 LOSS FUNCTIONS AND DYNAMIC WEIGHTING STRATEGIES

The loss function design systematically integrates established techniques to address acupoint-specific multi-task learning challenges: uncertainty-based weighting Kendall et al. (2018), dynamic visibility masking, and task-specific loss functions. This engineering approach enables adaptive task balancing while maintaining optimal convergence characteristics for the domain.

### 3.5.1 PER-MERIDIAN UNCERTAINTY WEIGHTING

Following the standard multi-task learning approach of Kendall et al. (2018), we employ learnable uncertainty parameters to automatically balance task weights. Our implementation adapts this framework to meridian-specific processing, where each of the nine meridian layers maintains independent uncertainty parameters for multi-tasks, as shown in Equation 1:

$$L_{total} = \sum_{i=1}^{9} \left( \frac{L_v^{(i)}}{2(\sigma_v^{(i)})^2} + \frac{L_c^{(i)}}{2(\sigma_c^{(i)})^2} + \frac{L_n^{(i)}}{2(\sigma_n^{(i)})^2} + \log\left(\sigma_v^{(i)}\sigma_c^{(i)}\sigma_n^{(i)}\right) \right) \tag{1}$$

where $\sigma_v^{(i)} = e^{\log\_var\_v^{(i)}}$, $\sigma_c^{(i)} = e^{\log\_var\_c^{(i)}}$, $\sigma_n^{(i)} = e^{\log\_var\_n^{(i)}}$ represent learned uncertainty parameters for visibility, coordinate, and classification tasks in meridian $i$. Parameters are initialized to 0.0, 1.0, and -5.0 respectively based on empirical convergence patterns. The detailed probabilistic derivations and gradient analysis are presented in Appendix A.1.

### 3.5.2 DYNAMIC SOFT VISIBILITY MASK PADDING

The dynamic soft visibility mask padding mechanism represents a key engineering refinement that addresses the challenge of task coordination in multi-task learning for acupoint detection. Traditional approaches often use hard thresholding to mask losses for occluded acupoints, but this can lead to gradient discontinuities and suboptimal task interaction during training.

The soft masking mechanism computes dynamic weights based on visibility prediction confidence. For each acupoint $j$ in meridian $i$ (where $i \in \{1, ..., 9\}$ indexes meridians and $j \in \{1, ..., N_i\}$ indexes acupoints within meridian $i$), the mask weight is computed as Equation 2:

$$w_{mask}^{(i,j)} = \max\left(0, \min\left(1, \frac{p_{vis}^{(i,j)} - \tau_{low}}{\tau_{high} - \tau_{low}}\right)\right) \tag{2}$$

where $p_{vis}^{(i,j)}$ is the visibility probability from the visibility prediction head, and $\tau_{low} = 0.075$ and $\tau_{high} = 0.925$ define the uncertainty region. The formulation ensures smooth transitions between different confidence levels, avoiding gradient discontinuities that could destabilize training.

The mechanism operates in three distinct regimes. For high confidence visible predictions ($p_{vis}^{(i,j)} \geq 0.925$), the weight approaches 1.0, indicating full confidence in the visibility assessment. For high confidence occluded predictions ($p_{vis}^{(i,j)} < 0.075$), the weight approaches 0.0, effectively masking the coordinate and classification losses. In the uncertainty region ($0.075 \leq p_{vis}^{(i,j)} < 0.925$), the weight varies linearly, allowing the model to maintain gradient flow while reducing the influence of uncertain predictions.

An additional protective mechanism prevents complete gradient elimination in the uncertainty region through minimum weight clamping as $w_{mask}^{(i,j)} = \max(w_{mask}^{(i,j)}, 0.075)$.

This ensures that at least 7.5% of the gradient signal is preserved even for highly uncertain predictions, preventing gradient vanishing while maintaining the model's ability to learn from challenges.

### 3.5.3 Decaying Weight Function for False Positive Suppression

The decaying weight function addresses a critical challenge in coordinate regression: preventing the model from over-optimizing on false positive acupoints that should converge to the origin (0,0) but are incorrectly predicted as visible. Without this mechanism, the model tends to focus excessively on these "easy" regression targets, leading to negative convergence for more challenging visible acupoint localization. The decaying weight function is formulated as Equation 3:

$$
w_{\hat{t}}^{(i,j)} = \begin{cases} 1 & \text{if epoch} \leq 1 \vee \text{coord}_{\text{gt}}^{(i,j)} \neq (0,0) \\ w_{\min} + (1 - w_{\min}) \cdot \frac{1 - \tanh\left(\frac{4}{b-a}\left(\text{epoch} - \frac{a+b}{2}\right)\right)}{2} & \text{if epoch} > 1 \wedge \text{coord}_{\text{gt}}^{(i,j)} = (0,0) \end{cases} \tag{3}
$$

The function maintains full weight (1.0) for all visible acupoints and during the initial training phase, but gradually reduces the weight for acupoints with ground truth coordinates at the origin. The tanh-based decay function provides smooth weight reduction over a configurable epoch range, with parameters $a = 25$, $b = 125$, and $w_{min} = 0.03$ in our implementation.

This strategy encourages the model to prioritize learning accurate localization for visible acupoints while gradually reducing emphasis on the trivial task of predicting (0,0) for occluded points. The smooth decay prevents training instabilities that could arise from abrupt weight changes while ensuring that the model maintains some ability to handle false negative cases throughout training.

### 3.5.4 Task-Specific Loss Functions

The framework employs three specialized loss functions within the uncertainty weighting scheme, each tailored to the specific characteristics of visibility classification, coordinate regression, and acupoint classification tasks. These individual task losses are combined within each meridian layer to form the complete multi-task objective.

Visibility Classification Loss. The visibility classification task employs Binary Focal Loss Lin et al. (2017) to address the severe class imbalance inherent in acupuncture acupoint visibility. For meridian $i$, the visibility loss aggregates across all $N_i$ acupoints as shown in Equation 4:

$$
L_v^{(i)} = \sum_{j=1}^{N_i} \text{BinaryFocalLoss}(vis_{\text{pred}}^{(i,j)}, vis_{\text{gt}}^{(i,j)}) \tag{4}
$$

where $N_i$ represents the maximum number of acupoints in meridian $i$. The Binary Focal Loss for each acupoint $j$ is defined as $\text{BinaryFocalLoss}(p, y) = -\alpha_t(1 - p_t)^\gamma \log(p_t)$, with $p_t = p$ if $y = 1$ and $p_t = 1 - p$ if $y = 0$, where $\alpha_t = 0.25$ and $\gamma = 2.0$.

Coordinate Regression Loss. Coordinate regression employs Soft Wing Loss Feng et al. (2018), building upon Wing Loss foundations for robust acupoint localization as Equation 5:

$$
L_c^{(i)} = \sum_{j=1}^{N_i} w_{\hat{t}}^{(i,j)} \cdot w_{\text{mask}}^{(i,j)} \cdot \text{SoftWingLoss}(coord_{\text{pred}}^{(i,j)}, coord_{\text{gt}}^{(i,j)}) \tag{5}
$$

where the Soft Wing Loss is defined as $\text{SoftWing}(\Delta) = \begin{cases} \omega \log(1 + \frac{|\Delta|}{\epsilon}) & \text{if } |\Delta| < \theta \\ A \cdot |\Delta| - C & \text{if } |\Delta| \geq \theta \end{cases}$, with parameters $\omega = 4.0$, $\epsilon = 0.2$, $\theta = 0.5$, $A = \omega/\epsilon$, and $C = A\theta - \omega \log(1 + \theta/\epsilon)$, incorporating both dynamic weights $w_{\hat{t}}^{(i,j)}$ and visibility masks $w_{\text{mask}}^{(i,j)}$.

Classification Loss. Acupoint classification utilizes standard cross-entropy loss within each meridian-specific context, as shown in Equation 6:

$$
L_n^{(i)} = \sum_{j=1}^{N_i} w_{\text{mask}}^{(i,j)} \cdot \text{CrossEntropy}(name_{\text{pred}}^{(i,j)}, name_{\text{gt}}^{(i,j)}) \tag{6}
$$

where the cross-entropy is computed over meridian-specific acupoint classes, modulated by visibility-based weighting.

| Backbone NN | | Feature Dimention | Params (M) | Batch Size | Batch/s | Total Loss | Top Vis Acc. (%) | Regression Loss (pixel) |
|---|---|---|---|---|---|---|---|---|
| *ResNet* He et al. (2016) | ResNet 50 | 2048 | 167.09 | 16 | 5.64 | 8.49 | 94.39 | 0.89 |
| | | | | 36 | 2.63 | 511.73 | 76.49 | 1.74 |
| | ResNet 101 | | 224.20 | 16 | 3.23 | 9.46 | 93.69 | 0.91 |
| | | | | 36 | 1.90 | 514.74 | 76.58 | 1.71 |
| | ResNet 152 | | 271.27 | 16 | 3.26 | 8.47 | 94.47 | 0.89 |
| | | | | 36 | 1.42 | 512.37 | 77.09 | 1.70 |
| *DenseNet* Huang et al. (2017) | DenseNet 169 | 1664 | 123.71 | 16 | 4.12 | 7.91 | 95.17 | 0.88 |
| | | | | 36 | 1.92 | 516.76 | 76.51 | 1.73 |
| | DenseNet 201 | 1920 | 147.74 | 16 | 3.40 | 7.73 | 95.30 | 0.87 |
| | | | | 36 | 1.53 | 512.81 | 76.71 | 1.70 |
| *ConvNeXt* Liu et al. (2022) | ConvNeXt Base | 1024 | 335.95 | 16 | 1.59 | 8.06 | 95.70 | 0.89 |
| | | | | 36 | 0.63 | 518.73 | 76.42 | 1.74 |
| | ConvNeXt Large | 1536 | 671.1 | 16 | 0.35 | 7.14 | 95.93 | 0.86 |
| *VGG Net* Simonyan & Zisserman (2014) | VGG 16 | 4096 | 555.84 | 16 | 3.63 | 7.29 | 95.23 | 0.85 |
| | VGG 16 BN | | 555.89 | 36 | 1.77 | 496.84 | 81.27 | 1.62 |
| | | | | | 1.41 | 500.72 | 78.83 | 1.68 |
| | VGG 19 | | 571.77 | 16 | 3.2 | 7.73 | 94.77 | 0.86 |
| | VGG 19 BN | | 571.83 | 36 | 1.54 | 488.69 | 82.40 | 1.60 |
| | | | | | 1.25 | 505.77 | 80.63 | 1.64 |

Table 1: Backbone neural network ablation study results (with 20% dataset) comparing different architectures across multiple performance metrics.

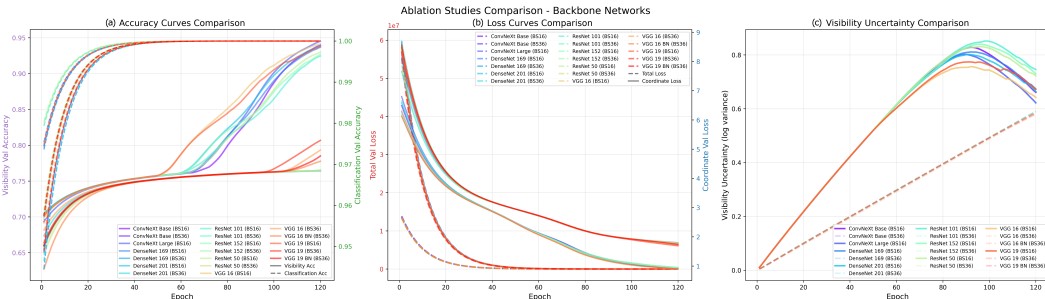

Figure 2: Backbone neural network ablation study results (with 20% dataset) showing the comparison in accuracy (a), loss (b) and uncertainty (c) between 19 tasks.

## 4 EXPERIMENTAL RESULTS AND VALIDATION

### 4.1 TRAINING CONFIGURATION AND BENCHMARK RESULTS

Experiments were conducted on NVIDIA RTX6000ADA GPU using PyTorch 2.4.1 with CUDA 12.4. Training used 512×512 images, batch size 16 and 36, learning rate $5 \times 10^{-5}$ (AdamW), weight decay $1 \times 10^{-5}$, and 150 epochs. Soft visibility masking parameters were $\tau_{\text{low}} = 0.075$, $\tau_{\text{high}} = 0.925$, with decaying weight function using a_epoch = 25, b_epoch = 125, and w_min = 0.03.

Figure 4 shows training dynamics across three aspects. Visibility classification achieved 99.97% peak accuracy, reaching 90% at epoch 39 and 99% at epoch 86, demonstrating rapid convergence. Total validation loss decreased to -74 by epoch 62, while coordinate regression loss reached 0.62 pixels at epoch 39, achieving sub-pixel accuracy. The uncertainty parameters evolved as follows: coordinate regression uncertainty stabilized at 3.2 after 100 epochs, visibility classification uncertainty peaked at 1.0 (epoch 50) then decreased to -1.0 (epoch 150), and acupoint classification uncertainty decreased from -5.0 to -11.0 after epoch 50, indicating progressive task simplification.

The model achieved exceptional error suppression with 0% false positive rate and 0.7‰ false negative rate after applying the decaying weight mechanism, demonstrating the effectiveness of combined explicit and implicit visibility learning for clinical applications.

### 4.2 BACKBONE NEURAL NETWORK ABLATION STUDY

We evaluated four neural network families (ResNet He et al. (2016), DenseNet Huang et al. (2017), ConvNeXt Liu et al. (2022), and VGG Simonyan & Zisserman (2014)) across eleven configurations and nineteen training sessions using 20% of the dataset. Table 1 summarizes the quantitative results across different architectures, presenting feature dimensions, parameter counts, batch sizes, throughput rates, and performance metrics. Two batch sizes (16 and 36) were tested to assess accuracy-efficiency trade-offs.

| Experimental Configuration | Module Combination | Top Val Vis Acc (%) | Min Val Coord Loss (Pixels) | Coord Loss (mm) | False Positive Rate (%) | e 90 | e 95 |
|---|---|---|---|---|---|---|---|
| *A0: Maintain benchmark configuration* | *A0 B0 C0* | *95.30* | *0.8759* | *0.4735* | *6.14* | *92* | *118* |
| A1: Random Assignment (Structure Preserved) | A1 B0 C0 | 94.89 | 0.7826 | 0.4230 | 7.04 | 98 | - |
| A2: Random Assignment (Balanced) | A2 B0 C0 | 94.02 | 0.8306 | 0.4490 | 6.72 | 98 | - |
| A3: Unified prediction for all 174 acupoints Sun et al. (2025) | - | 99.73 (3000 epoch) | 0.7560 | 0.4086 | - | 1305 | - |
| *B0: Maintain benchmark configuration* | *A0 B0 C0* | *95.30* | *0.8759* | *0.4735* | *6.14* | *92* | *118* |
| B1: Hard binary thresholding (p_vis $\geq$ 0.5) | A0 B1 C0 | 85.25 | 0.9402 | 0.5082 | 19.61 | - | - |
| B2: No visibility masking for multi-tasks Sun et al. (2025) | - | 99.73 (3000 epoch) | 0.7560 | 0.4086 | - | 1305 | - |
| *C0: Maintain benchmark configuration* | *A0 B0 C0* | *95.30* | *0.8759* | *0.4735* | *6.14* | *92* | *118* |
| C1: Fixed initialization at zero ($\sigma = 1$) | A0 B0 C1 | 94.23 | 1.3178 | 0.7123 | 6.80 | 88 | - |
| C2: No uncertainty parameters Sun et al. (2025) | - | 99.73 (3000 epoch) | 0.7560 | 0.4086 | - | 1305 | - |
| *D0: With Visibility Result Guide* | *A0 B0 C0 D0* | *95.30* | *0.8759* | *0.4735* | *6.14* | *92* | *118* |
| D1: Without Visibility Result Guide | A1 B0 C0 D1 | 80.33 / 84.22 (Final) | 0.3720 / 1.9945 (Final) | 1.0781 | 34.71 | - | - |
| A0 B1 C1 | | 84.96 | 0.9488 | 0.5129 | 18.12 | - | - |
| A1 B0 C1 | | 83.17 | 0.8695 | 0.4700 | 19.57 | - | - |
| A1 B1 C0 | | 92.59 | 0.8272 | 0.4471 | 6.40 | 95 | - |
| A1 B1 C1 | | 84.31 | 0.8511 | 0.4601 | 14.98 | - | - |
| A2 B0 C1 | | 82.35 | 0.8940 | 0.4832 | 19.34 | - | - |
| A2 B1 C0 | | 92.79 | 0.8544 | 0.4618 | 9.09 | 98 | - |
| A2 B1 C1 | | 83.11 | 0.8900 | 0.4811 | 18.81 | - | - |

Table 2: Module ablation study results (with 20% dataset) comparing different combinations across multiple performance metrics.

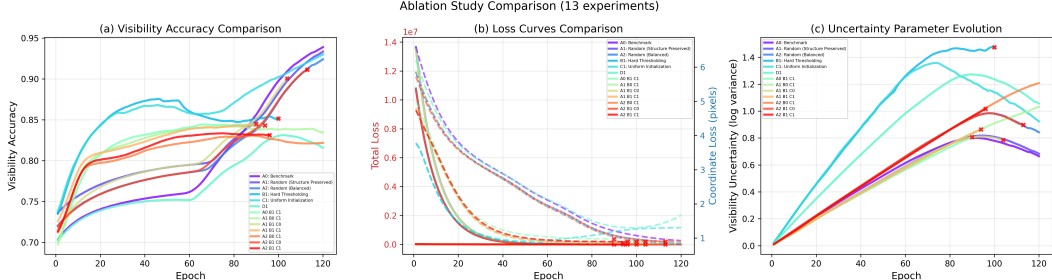

Figure 3: Module ablation study results (with 20% dataset) showing the comparison in accuracy (a), loss (b) and uncertainty (c) between 13 combinations.

Figure 2 shows comprehensive training dynamics across three dimensions: validation accuracy curves (subplot a), loss comparisons (subplot b), and uncertainty parameter evolution (subplot c). Batch size 16 configurations (B16) significantly outperformed batch size 36 (B36) with accuracy ranges of 93.60%-95.93% versus 76.42%-82.40%. As illustrated in Figure 2(a), B16 demonstrated accelerated gains between epochs 50-75, while B36 required 50 additional epochs for similar convergence, indicating larger batch sizes impede multi-task learning efficiency. The loss curves in Figure 2(b) and uncertainty dynamics in Figure 2(c) further confirm these patterns. ConvNeXt Large achieved the highest accuracy (95.93% as shown in Table 1), but DenseNet201 was selected as optimal backbone, balancing performance (95.30%) and efficiency (147.74M parameters, 3.40 batch/s) compared to ConvNeXt Large (671.1M parameters, 0.35 batch/s). This enables efficient clinical deployment while maintaining medical-grade precision.

## 4.3 NEURAL NETWORK COMPONENTS ABLATION STUDY

We conducted comprehensive ablation studies across four design dimensions on 20% of the dataset, **explicitly comparing our method against the baseline model acuSim Sun et al. (2025).** Table 2 presents quantitative results for individual module variations and combinations alongside this baseline. Figure 3 illustrates training dynamics of thirteen configurations: accuracy evolution (a), loss curves (b), and uncertainty parameters (c).

**Ablation A.** The meridian layering ablation experiments (A0, A1, A2, A3) demonstrate the critical importance of medical prior-based organization. As shown in both Figure 5(a) and Table 2, the medical prior-based configuration (A0, purple curve in Figure 5(a)) achieves optimal final accuracy (95.30%) despite initially trailing random assignments (A1: 94.89%, A2: 94.02%) during the first 65 epochs, exhibiting slower early convergence but significantly higher acceleration thereafter. A0 surpasses A2 at epoch 76 and A1 at epoch 82, maintaining superiority through remaining training. Notably, A2 consistently underperforms A1 throughout training, indicating that meridian structure preservation matters even without semantic correctness. The baseline Sun et al. (2025) shows substantially degraded performance, confirming the value of layered architecture itself.

**Ablation B.** The visibility encoding experiments (B0, B1, B2) validate soft masking superiority, as evidenced in Table 2 and Figure 3. Hard thresholding (B1) with fixed 0.5 threshold causes gradient explosion, terminating training due to abrupt switching between full gradient propagation and complete masking at the 0.5 decision boundary. In contrast, removing visibility masking entirely (B2, baseline Sun et al. (2025)) exhibits pathological training dynamics characterized by extremely slow convergence. Without visibility guidance, the network prioritizes learning trivial patterns for occluded acupoints clustered at origin (0,0), causing negative convergence for visible acupoints where the coordinate regression task struggles to make meaningful progress. This phenomenon mirrors the issues observed in *Ablation D*, demonstrating that visibility-based task modulation is essential regardless of implementation approach. The soft masking approach (B0) maintains stable convergence by providing smooth gradient transitions in the uncertainty region ($\tau_{low}$ to $\tau_{high}$), preventing both gradient explosion and negative convergence patterns.

**Ablation C.** The uncertainty parameter study (C0 vs C1) confirms the importance of empirical initialization. The benchmark configuration (C0) uses initial values 1.0, 0.0, and -5.0 for coordinate, visibility, and classification tasks respectively, achieving stable dynamics as illustrated in Figure 3(c). Uniform initialization (C1) with all parameters at 0.0 shows suboptimal convergence with delayed peaks (approximately 20 epochs later) and slower overall convergence (requiring 30% more epochs to reach 95% accuracy), achieving only 94.23% accuracy with significantly higher coordinate loss (1.3178 pixels vs 0.8759 pixels for C0, as reported in Table 2). This performance degradation occurs because uniform initialization assigns equal initial weights to all tasks, forcing the network to discover optimal task prioritization from scratch.

**Ablation D.** The visibility guidance ablation (D0 vs D1) validates the critical role of visibility-aware task coordination beyond simple confidence weighting. While D1 maintains visibility prediction training, it removes visibility guidance for coordinate regression and classification tasks by setting all visibility masks to 1. As shown in Table 2, D1 achieves only 80.33% peak accuracy with substantially elevated false positive rate (34.71%) compared to configurations with similar accuracy levels. The training dynamics in Figure 3(b) reveal characteristic overfitting patterns: the loss curve (cyan) exhibits rapid initial decrease followed by anomalous increase, indicating the network overfits to trivially locating invisible acupoints at origin (0,0) rather than learning meaningful spatial patterns for visible points. This coordinate task dominance, reflected in the uncertainty parameter evolution (Figure 3(c)), disrupts classification learning for invisible points and propagates instability back to visibility prediction through multi-task coupling, resulting in degraded accuracy and abnormally high false positives. These pathological dynamics demonstrate that visibility guidance provides essential task-specific supervision rather than merely confidence-based reweighting, preventing the network from exploiting trivial solutions and maintaining balanced multi-task learning.

## 5 CONCLUSION AND FUTURE WORK

This paper introduces acuSimNet, a domain-driven system-level engineering framework applying hierarchical multi-task learning to multi-view, self-occlusion-aware acupoint visibility prediction. We address three fundamental limitations: acupoint localization constrained to single viewpoints, human landmark detection assuming perpetual visibility, and training instability in existing acuSim work Sun et al. (2025). Our engineering contributions in medical prior-based meridian layering, dynamic soft visibility encoding, and systematic integration of uncertainty-based task weighting Kendall et al. (2018), incorporate TCM prior, enable robust occlusion handling, and optimize inter-task scheduling through careful adaptation of established techniques.

Experimental results demonstrate 99.97% validation accuracy in 86 epochs, achieving 35× convergence acceleration versus baseline Sun et al. (2025). Ablation studies validate each component: hierarchical meridian structure prevents training instability, soft visibility masking maintains gradient stability, and empirical uncertainty initialization outperforms uniform approaches. DenseNet201 provides optimal balance (95.30% accuracy, 147.74M parameters, 3.40 batch/s). Clinical relevance is demonstrated by 0% false positive and 0.7‰ false negative rates.

While achieving strong results on synthetic acuSim data, preliminary evaluation on real clinical images reveals domain gap challenges in generalization capabilities. Therefore, future directions include transfer learning to real clinical images addressing domain gaps (lighting, skin variations, occlusion patterns), model distillation for edge deployment, and AR/VR integration enabling real-time multi-view visualization for clinical procedures and TCM education. These advances would bridge traditional medical knowledge with modern computational capabilities.

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

# A APPENDIX

## A.1 UNCERTAINTY PARAMETER DERIVATION AND THEORETICAL FOUNDATION

The uncertainty weighting mechanism in acuSimNet is derived from Bayesian deep learning principles, where task-specific uncertainty quantification enables automatic balancing of multi-task objectives. This section provides the complete theoretical derivation and gradient analysis for the learnable uncertainty parameters.

### A.1.1 PROBABILISTIC FOUNDATION

The uncertainty parameters represent learned estimates of task-specific noise levels, formulated within a probabilistic framework where each task's likelihood is modeled as a Gaussian distribution with learned variance. For a given task with predictions $\hat{y}$ and ground truth $y$, the likelihood is:

$$p(y|\hat{y}, \sigma^2) = \frac{1}{\sqrt{2\pi\sigma^2}} \exp\left(-\frac{||y - \hat{y}||^2}{2\sigma^2}\right) \tag{7}$$

Taking the negative log-likelihood and ignoring constant terms yields the weighted loss formulation:

$$-\log p(y|\hat{y}, \sigma^2) \propto \frac{||y - \hat{y}||^2}{2\sigma^2} + \log \sigma \tag{8}$$

This formulation naturally balances the data fitting term (weighted by $1/\sigma^2$) with a regularization term ($\log \sigma$) that prevents the variance from growing unbounded.

### A.1.2 PARAMETERIZATION AND CONSTRAINTS

To ensure numerical stability and positive variance values, the actual variance is parameterized using the exponential of learnable log-variance parameters:

$$\sigma^2 = e^{2 \cdot \text{log\_var}} \tag{9}$$

This parameterization guarantees $\sigma^2 > 0$ for any real-valued log_var, while providing stable gradients through the exponential function. The factor of 2 in the exponent ensures that log_var directly corresponds to the logarithm of the standard deviation rather than the variance.

### A.1.3 GRADIENT ANALYSIS AND LEARNING DYNAMICS

The gradient of the total loss with respect to each log-variance parameter reveals the automatic balancing mechanism. For the visibility task in meridian $i$:

$$\frac{\partial L_{\text{total}}}{\partial \text{log\_var\_v}^{(i)}} = \frac{\partial}{\partial \text{log\_var\_v}^{(i)}} \left(\frac{L_v^{(i)}}{2e^{2\text{log\_var\_v}^{(i)}}} + \text{log\_var\_v}^{(i)}\right) \tag{10}$$

Computing the partial derivative:

$$\frac{\partial L_{\text{total}}}{\partial \text{log\_var\_v}^{(i)}} = -\frac{L_v^{(i)}}{e^{2\text{log\_var\_v}^{(i)}}} + 1 = -\frac{L_v^{(i)}}{(\sigma_v^{(i)})^2} + 1 \tag{11}$$

This gradient expression reveals the automatic balancing mechanism:

Case 1: High Task Loss ($L_v^{(i)} > (\sigma_v^{(i)})^2$): The gradient becomes negative, causing log_var_v$^{(i)}$ to decrease during optimization. This reduces $\sigma_v^{(i)}$, thereby increasing the task weight $\frac{1}{2(\sigma_v^{(i)})^2}$, giving more emphasis to the difficult task.

Case 2: Low Task Loss ($L_v^{(i)} < (\sigma_v^{(i)})^2$): The gradient becomes positive, causing log_var_v$^{(i)}$ to increase during optimization. This increases $\sigma_v^{(i)}$, thereby decreasing the task weight $\frac{1}{2(\sigma_v^{(i)})^2}$, reducing emphasis on the easy task.

Case 3: Balanced State ($L_v^{(i)} = (\sigma_v^{(i)})^2$): The gradient equals zero, indicating an equilibrium where the task weight is optimally balanced relative to the current loss magnitude.

### A.1.4 MULTI-TASK INTERACTION AND CONVERGENCE PROPERTIES

The uncertainty weighting mechanism facilitates automatic task prioritization that adapts throughout training. During early training phases, tasks with higher initial losses receive increased attention through reduced uncertainty parameters. As training progresses and task losses decrease, the uncertainty parameters adjust to maintain optimal balance.

The regularization term $\log(\sigma_v^{(i)} \sigma_c^{(i)} \sigma_n^{(i)})$ serves multiple purposes:

1. Variance Regularization: Prevents uncertainty parameters from growing unbounded, which would effectively eliminate task contributions.

2. Task Balance Maintenance: Encourages the model to maintain meaningful contributions from all tasks rather than completely ignoring difficult ones.

3. Numerical Stability: Ensures that the optimization landscape remains well-conditioned throughout training.

The initialization strategy leverages empirical observations of task convergence patterns. Classification tasks typically exhibit higher initial losses due to the discrete nature of the prediction space and higher-dimensional output requirements. Coordinate regression tasks often converge more rapidly for visible acupoints due to the continuous prediction space and direct spatial relationships. Visibility classification represents an intermediate case, requiring moderate initial weighting to establish reliable occlusion assessment before other tasks can benefit from dynamic masking.

### A.1.5 IMPLEMENTATION CONSIDERATIONS

The uncertainty parameters are implemented as standard PyTorch parameters within each meridian layer, participating in the standard gradient-based optimization process alongside other model weights. The parameter tensor has shape $[9, 3]$ corresponding to nine meridians and three tasks per meridian.

During the forward pass, the uncertainty parameters are converted to actual variances using the exponential function, and the weighted losses are computed according to the formulation above. The backward pass automatically computes gradients for all uncertainty parameters through standard automatic differentiation, enabling seamless integration with existing optimization algorithms.

### A.2 SUPPLEMENTARY FIGURES AND TABLES

| Methodology | Target Area | Subjects | Acupoints | Image | Total Annotations | Image Resolution | Multi-view | Visibility Info |
|---|---|---|---|---|---|---|---|---|
| Sun et al. (2025) | Cervicocranial | 504 | 174 | 63,936 | 11,126,952 | 1024*1024 | Yes | Yes |
| Ji & Zhou | Back | - | 4 | - | - | - | No | No |
| Masood & Qi (2022) | Human Hands | - | 9 | 1,200 | 28,800 | 848*480 | Yes | No |
| Sun et al. (2020) | Forearm | - | 2 | 719 | 2,876 | - | No | No |
| Zhang et al. (2023) | Facial | 4 | 43 | 654 | 28,122 | - | No | No |

Table 3: Comparison between existing acupoint datasets and synthetic approaches

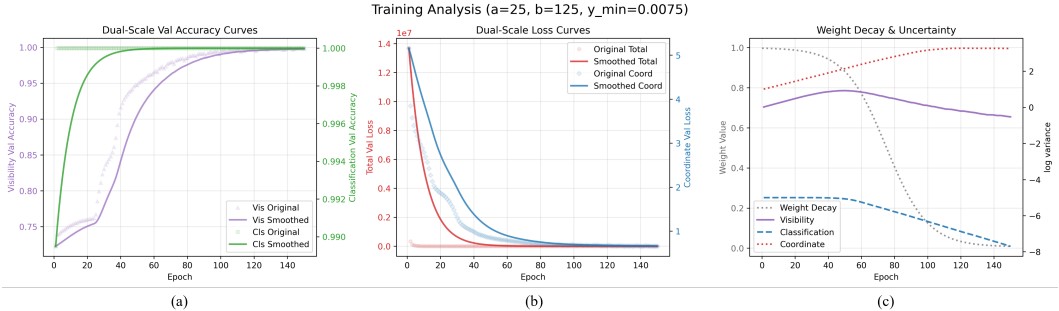

Figure 4: Benchmark training plot showing the accuracy (a), loss (b) and uncertainty (c) results.

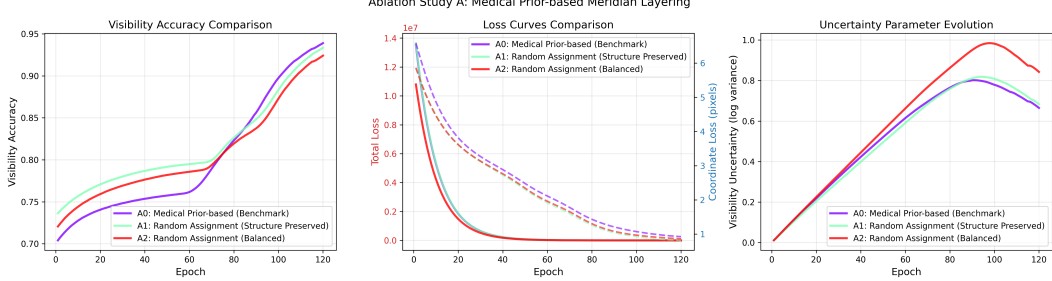

Figure 5: Ablation study results (with 20% dataset) showing the comparison in accuracy (a), loss (b) and uncertainty (c) between A0, A1 and A2.

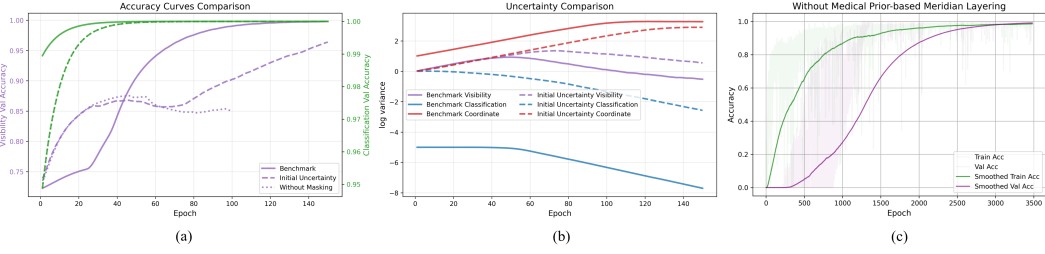

Figure 6: Component ablation study results (with full dataset) showing visibility accuracy comparisons, uncertainty parameter evolution, and training stability analysis

