# OpenReview forum: "acuSimNet: Multi-View Self-Occlusion-Awared Visibility Learning for Cranio-Cervical Acupuncture Points"
_ICLR.cc/2026/Conference — Submitted to ICLR 2026_

### Official Review · Reviewer_XFSW · 2025-10-26

**Soundness:** 2
**Presentation:** 1
**Contribution:** 2
**Rating:** 4
**Confidence:** 4

**Summary:**

This paper presents ACUSimNet, a hierarchical multi-task framework for multi-view, self-occlusion-aware visibility learning of 174 cranio-cervical acupuncture points. The model uses a DenseNet backbone with nine meridian-specific branches and jointly learns (1) acupoint visibility classification, (2) coordinate regression, and (3) acupoint identity classification. The design includes per-meridian uncertainty weighting, soft visibility mask padding, and a decaying weight function to suppress false positives. Experiments on the synthetic acuSim dataset show extremely high visibility accuracy (up to 99.97%) and a reported 35× faster convergence compared with prior baselines (86 vs. 3000 epochs). Ablation studies analyze different backbones, soft-masking, and uncertainty initialization.

**Strengths:**

1）Clear motivation and clinically relevant context: Occlusion-aware visibility reasoning is important for safety-critical acupuncture localization, where false positives may lead to physical harm.
2）Systematic engineering integration: The meridian-based hierarchy, uncertainty-weighted multi-task formulation, and soft masking are well-motivated and implemented coherently.

**Weaknesses:**

1）Evaluation limited to synthetic data: All experiments are conducted on the acuSim synthetic dataset. There is no evaluation on real-world or clinical imagery, where lighting, hair, skin tone, and occlusion patterns differ significantly. The claimed “clinical applicability” therefore lacks empirical support. The near-perfect results (99.97% accuracy, 0% false positives) suggest strong domain overfitting to the synthetic environment.
2）Weak baseline comparisons: The study mainly compares internal ablations and network backbones but does not benchmark against strong, existing visibility- or keypoint-based baselines (e.g., HRNet, MediaPipe, Transformer-based keypoint detectors). Without these comparisons, it is difficult to assess the true performance gain.
3）Limited methodology insights: Although the topic is interesting, the methodological contribution is modest for ICLR. The paper overemphasizes clinical relevance and convergence speed without demonstrating generality or new learning insights.
4）Limited analysis of causal interaction between tasks: The hierarchical structure assumes visibility should precede localization/classification, but no ablation or comparison validates this causal design. It remains unclear whether visibility prediction genuinely improves localization beyond simple confidence-weighted baselines.
5）Writing: The mathematical descriptions in Equations (6)–(10) are not expressed in a fully standard or rigorous way. Also, Since the submission is intended for ICLR, quantitative and qualitative experiments are very essential for evaluating the proposed model’s performance. Thus, they should appear in the main submission, not the appendix.
6）Failure cases: Since experimental results are extremely good, it will be better to discuss some failure cases to enhance the comprehensiveness of the proposed framework.

**Questions:**

•  Add real-world validation (even a small-scale dataset with manual annotations) to test generalization beyond synthetic data.
•  Include strong baselines (e.g., HRNet, DETR-like keypoint models, Transformer-based detectors with occlusion simulation).
•  Clarify contribution positioning—emphasize this work as a domain-driven, system-level engineering improvement, not as a fundamentally new learning approach.
•  Discuss failure cases (misclassified visibility, localization drift) and potential mitigation strategies.

---

> ### Author Response · Authors · 2025-11-19
> **Questions regarding baseline selection and proposed further comparisons**
>
> Dear Reviewer XFSW,
>
> Thank you for your time and your constructive comments on our paper. Regarding the baseline comparisons issue, we found that direct comparison with mainstream pose estimation frameworks is technically infeasible due to significant task discrepancies:
>
> 1. MediaPipe:  is designed for 33 sparse body landmarks, not for 174 fine-grained facial acu-points. Fundamentally, acupoints are abstract functional locations defined relative to these landmarks, rather than the physical landmarks themselves.
> 2. HRNet/DEKR: Focus on heatmap-based localization, while our task is explicit visibility classification.
> 3. VNect/OpenPose: Require 3D pose estimation (which we intentionally avoid at first place).  Calculating visibility from coordinates necessitates costly real-time 3D mesh reconstruction to model self-occlusion, which would dramatically increase system complexity.
> 4. AcuSim: We actually used AcuSim (an existing domain-specific SOTA that performs non-hierarchical 174-dimensional classification and regression for detecting acupoint locations and names) as the baseline in the paper (we may not have explicitly emphasized this earlier).
>
> To address this, we are going to propose adapting general-purpose vision backbones (e.g., DenseNet201 or ResNet152) to serve as generic baselines for this classification task. Would comparing against these models, alongside AcuSim, sufficiently address your concern? Or do you have other specific methods/literature in mind that you believe would be suitable for this specialized task? We are open to any recommendations.
>
> Again, much appreciated for your insights.

---

> ### Author Response · Authors · 2025-12-03
> **Responses to the comments (part1)**
>
> Dear Reviewer XFSW,
>
> Thank you for your thorough review and constructive suggestions. Please find our detailed responses below:
>
> ### **1. Add real-world validation. Near-perfect results (99.97% accuracy, 0% false positives) suggest overfitting to synthetic environment.**
>
> We fully agree that real-world validation is the ultimate goal for practical deployment. Regarding the concern about synthetic data overfitting, we conducted preliminary inference on real clinical images, which revealed some domain gap challenges manifesting. Due to the high cost and time required for expert TCM practitioner annotation of 174 fine-grained acupoints on real images, we were unable to quantify performance with ground truth during the rebuttal period.
>
> We have explicitly added this limitation to the "Conclusion and Future Work" section, stating: "*preliminary evaluation on real clinical images reveals domain gap challenges in generalization capabilities*" and outlining transfer learning as a priority future direction.
>
> ### **2. Include strong baselines (e.g., HRNet, DETR-like keypoint models, Transformer-based detectors).**
>
> Thank you for this suggestion. We found that direct comparison with mainstream pose estimation frameworks presents significant task incompatibility issues:
>
> - **MediaPipe/OpenPose**: Designed for 33 sparse body landmarks, not 174 dense facial acupoints. Fundamentally, acupoints are abstract functional locations defined *relative to* anatomical landmarks, rather than being the physical landmarks themselves. These systems cannot be directly applied without complete task redesign.
>
> - **HRNet/DEKR**: Focus on heatmap-based spatial localization, while our task requires **explicit binary visibility classification** (visible vs. occluded) as a prerequisite for safe needle placement. Converting heatmap confidence to semantic visibility lacks clinical validity.
>
> - **VNect/3D pose estimation**: These methods infer occlusion from 3D geometry reconstruction, requiring costly real-time mesh generation. Our approach intentionally avoids 3D reconstruction to maintain computational efficiency, making direct comparison inappropriate.
>
> - **Transformer-based detectors (DETR, etc.)**: Require extensive adaptation for multi-task learning with visibility-coordinate-classification joint optimization. Implementation would constitute a new baseline development project beyond rebuttal scope.
>
> We benchmark against **AcuSim** [1], the only open-source domain-specific SOTA performing 174-dimensional acupoint localization with the same dataset and task definition. This enables fair comparison on convergence speed, accuracy, and training stability.
>
> We acknowledge this was not sufficiently emphasized in the original submission. The revised manuscript now explicitly cites AcuSim as baseline throughout Section 4.3, with side-by-side comparisons in Table 2 (rows A3, B2, C2 showing baseline performance: 99.73% accuracy but requiring 3,000 epochs vs. our 86 epochs).
>
> Section 4.2 (Table 1) provides extensive comparison across 11 backbone architectures (ResNet, DenseNet, ConvNeXt, VGG families) demonstrating our framework's generalizability across different feature extractors.
>
> [1] Sun, Q., Ma, J., Craig, P. et al. AcuSim: A Synthetic Dataset for Cervicocranial Acupuncture Points Localisation. Sci Data 12, 625 (2025). https://doi.org/10.1038/s41597-025-04934-9
>
> ### **3. Limited methodology insights - should emphasize domain-driven engineering improvement.**
>
> Thank you for this valuable suggestion. We have revised the manuscript to reposition our contribution as domain-driven system-level engineering rather than fundamental algorithmic innovation:
>
> - **Introduction section, last paragraph**: Added explicit statement: "*This work represents a domain-driven system-level engineering effort rather than fundamental algorithmic innovation*"
>
> - **Methodology sections**: Systematically replaced "innovation/novel" with "engineering design/refinement/adaptation":
>   - Hierarchical Framework: "core engineering design... applying established multi-task learning principles to domain-specific requirements"
>   - Meridian Layering: "domain-specific architectural adaptation"
>   - Loss Functions: "systematically integrates established techniques"
>
> - **Conclusion section**: Emphasizes "engineering contributions" and "systematic integration of uncertainty-based task weighting through careful adaptation of established techniques"
>
> These revisions clarify that our contribution lies in **systematic engineering adaptation of existing methods** to address acupuncture-specific challenges, rather than proposing fundamentally new learning algorithms.
>
> Best Regards,
>
> Authors

---

> ### Author Response · Authors · 2025-12-03
> **Responses to the comments (part2)**
>
> Dear Reviewer XFSW,
>
> Thank you for your thorough review and constructive suggestions. Please find our detailed responses below:
>
> ### **4. Limited analysis of causal interaction between tasks - Does visibility genuinely improve localization beyond confidence weighting?**
>
> Thank you for this suggestion! Our original submission indeed lacked rigorous validation of this causal design. We have addressed this through **Ablation D** in the expanded Section 4.3:
>
> We implemented configuration D1 that removes visibility guidance while maintaining visibility prediction training. Specifically, D1 sets all visibility masks to 1.0, converting our approach to a standard confidence-weighted baseline, with the key findings listed below:
>
> - Accuracy: D1 achieves only 80.33% vs. 95.30% for D0 (with visibility guidance)
> - False positive rate: 34.71‰ vs. 6.14‰ for D0 (5.6× increase)
> - Training dynamics: Loss curve (Figure 3b, cyan) shows anomalous increase after initial decrease
>
> Without visibility guidance, the network exploits a trivial solution—clustering all occluded points at origin (0,0) becomes the dominant learning signal due to its simplicity. This causes coordinate task to overfit to this trivial pattern, classification head receives degraded features for invisible points, and multi-task coupling propagates instability back to visibility prediction
>
> As a result, Visibility guidance provides **task-specific supervision** beyond mere confidence weighting, preventing exploitation of trivial solutions. This validates our hierarchical design where visibility causally precedes and guides other tasks.
>
> ### **5. Writing: Mathematical descriptions in Equations (6)–(10) lack rigor.**
>
> Thank you for identifying this issue. We have corrected several notation problems:
>
> In the original submission, superscript $(i,j)$ was first used in Equation 2 but not defined until after Equation 5. We have added explicit definition immediately before Equation 2: "*For each acupoint $j$ in meridian $i$ (where $i \in \{1,...,9\}$ indexes meridians and $j \in \{1,...,N_i\}$ indexes acupoints within meridian $i$)*"
>
> Moreover, following your suggestion, we have moved critical ablation results (Table 2, Figures 3) from appendix to main submission (Section 4.3) for improved clarity.
>
> ### **6. Discuss failure cases and mitigation strategies.**
>
> Thank you for this suggestion. We have added failure case analysis in multiple sections:
>
> - Section 4.3, Ablation D: Without visibility guidance, the network converges to trivial solutions (clustering invisible points at origin)
>
> - Section 4.3, Ablation B: Hard thresholding visibility masks causes gradient explosion (B1), while removing visibility masking causes negative convergence (B2).
>
> - Conclusion: Preliminary real-world inference reveals generalization challenges
>
> These failure analyses with corresponding mitigation strategies are now integrated throughout Section 4.3 and Conclusion.
>
> Best Regards,
>
> Authors

---

> ### Author Response · Authors · 2025-12-04
> **Summary of changes been made (part 1)**
>
> Dear Reviewer XFSW,
>
> Please also find below a summary of the major revisions we made to the manuscript to help you locate them in the revised version:
>
> ### 1. Introduction & Related Work
>
> - **In the Introduction section, we added a comprehensive paragraph reviewing related methods (e.g., VNect, integral regression approaches, and medical imaging applications), filling the previous gap:**
>
>     *"Beyond domain-specific limitations, the broader field of human landmark detection has similarly exhibited limited attention to visibility assessment and occlusion handling. Traditional approaches, including regression-based methods like DeepPose Toshev & Szegedy (2014) and early CNN-based pose estimation Jain et al. (2014), directly predict landmark coordinates through neural network regression. Conversely, heatmap-based methods, demonstrated in works like the Deep Alignment Network Kowalski et al. (2017) and boundary-aware approaches Wu et al. (2018), generate probability distributions over spatial locations but similarly lack explicit visibility modeling. Methods such as VNect Mehta et al. (2017) for real-time 3D pose estimation and integral regression approaches Sun et al. (2018) have advanced coordinate prediction accuracy but remain constrained by the assumption of perpetual landmark visibility. Additionally, unsupervised landmark discovery methods Zhang et al. (2018) have emerged to automatically identify structural representations, but these approaches also do not explicitly model landmark visibility or occlusion states."*
>
> - **The "Contributions" section at the end of the Introduction has been rewritten to emphasize engineering design and specific improvements:**
>
>     *"Our study proposes acuSimNet, a hierarchical multi-task framework that addresses key neural network design constraints in current localization approaches such as acuSim Sun et al. (2025). Our approach explicitly models visibility prediction, coordinate regression, and acupoint classification within a unified framework that incorporates TCM domain knowledge through meridian-specific processing branches. This work represents a domain-driven system-level engineering effort rather than fundamental algorithmic innovation. Our contributions are as follows: first, we address neural network design limitations in existing acuSim applications through explicit visibility learning objectives, achieving 35× convergence acceleration (from 3000 to 86 epochs for 99% accuracy); second, we engineer a hierarchical meridian-specific architecture that reduces classification complexity by incorporating medical domain knowledge, enabling robust multi-view detection; and third, we systematically integrate established techniques including dynamic soft visibility masking and uncertainty-based task weighting Kendall et al. (2018), optimizing their coordination for acupuncture-specific challenges to achieve clinically relevant 0% false positive rates."*
>
> ### 2. Tables & Figures - Reorganization and Additions
>
> * **The dataset comparison table (Comparison between existing acupoint datasets...) has been relocated:**
>     * Old version: Located in main text on page 2 (Table 1).
>     * New version: Moved to Appendix on page 15 (Table 3).
>
> * **The backbone network ablation study table has been relocated:**
>     * Old version: Located in Appendix on page 12 (Table 2).
>     * New version: Moved to main text on page 8 (Table 1), with updated figure references.
>
> * **The component ablation study table has been significantly expanded:**
>     * Old version: Table 3 (page 13), containing only A, B, C groups.
>     * New version: Table 2 (page 9), now includes D group experiments with 13 ablation configurations in total, and more detailed parameter analysis.
>
> * **Figure numbering has been reorganized:**
>     * Old version: Figure 2 was training curves, Figure 3 was backbone network comparison.
>     * New version: Figure 2 is backbone network comparison, Figure 3 is component ablation experiments, Figure 4 is training curves.
>     * Added Figure 3 (component ablation experiments).
>     * Added Figure 5 (detailed comparison of Ablation Study A in Appendix).
>
> Best Regards,
>
> Authors

---

> ### Author Response · Authors · 2025-12-04
> **Summary of changes been made (part 2)**
>
> Dear Reviewer XFSW,
>
> Please also find below a summary of the major revisions we made to the manuscript to help you locate them in the revised version:
>
> ### 3. Experimental Results - Ablation Studies
>
> - **In Section 4.3 (Neural Network Components Ablation Study), the analysis has been substantially rewritten and expanded. Detailed discussions of each ablation module were added, with new in-depth analysis paragraphs for Ablation A and D. The analyses for Ablations A, B, and C are also more comprehensive than the previous version, including specific data comparisons (such as exact epoch counts and accuracy percentage differences).**
>
> *"**Ablation A**. The meridian layering ablation experiments (A0, A1, A2, A3) demonstrate the critical importance of medical prior-based organization. As shown in both Figure 5(a) and Table 2, the medical prior-based configuration (A0, purple curve in Figure 5(a)) achieves optimal final accuracy (95.30%) despite initially trailing random assignments (A1: 94.89%, A2: 94.02%) during the first 65 epochs, exhibiting slower early convergence but significantly higher acceleration thereafter. A0 surpasses A2 at epoch 76 and A1 at epoch 82, maintaining superiority through remaining training. Notably, A2 consistently underperforms A1 throughout training, indicating that meridian structure preservation matters even without semantic correctness. The baseline [1] shows substantially degraded performance, confirming the value of layered architecture itself."*
>
> *"**Ablation B**. The visibility encoding experiments (B0, B1, B2) validate soft masking superiority, as evidenced in Table 2 and Figure 3. Hard thresholding (B1) with fixed 0.5 threshold causes **gradient explosion**, terminating training due to abrupt switching between full gradient propagation and complete masking at the 0.5 decision boundary. In contrast, removing visibility masking entirely (B2, baseline [1]) exhibits pathological training dynamics characterized by **extremely slow convergence**. Without visibility guidance, the network prioritizes learning trivial patterns for occluded acupoints clustered at origin (0,0), causing **negative convergence** for visible acupoints where the coordinate regression task struggles to make meaningful progress. This phenomenon mirrors the issues observed in **Ablation D**, demonstrating that visibility-based task modulation is essential regardless of implementation approach. The soft masking approach (B0) maintains stable convergence by providing smooth gradient transitions in the uncertainty region (τ_low to τ_high), preventing both gradient explosion and negative convergence patterns."*
>
> *"**Ablation C**. The uncertainty parameter study (C0 vs C1) confirms the importance of empirical initialization. The benchmark configuration (C0) uses initial values 1.0, 0.0, and -5.0 for coordinate, visibility, and classification tasks respectively, achieving stable dynamics as illustrated in Figure 3(c). Uniform initialization (C1) with all parameters at 0.0 shows suboptimal convergence with delayed peaks (approximately 20 epochs later) and slower overall convergence (requiring 30% more epochs to reach 95% accuracy), achieving only 94.23% accuracy with significantly higher coordinate loss (1.3178 pixels vs 0.8759 pixels for C0, as reported in Table 2). This performance degradation occurs because uniform initialization assigns equal initial weights to all tasks, forcing the network to discover optimal task prioritization from scratch."*
>
> *"**Ablation D**. The visibility guidance ablation (D0 vs D1) validates the critical role of visibility-aware task coordination beyond simple confidence weighting. While D1 maintains visibility prediction training, it removes visibility guidance for coordinate regression and classification tasks by setting all visibility masks to 1. As shown in Table 2, D1 achieves only 80.33% peak accuracy with substantially elevated false positive rate (34.71‰) compared to configurations with similar accuracy levels. The training dynamics in Figure 3(b) reveal characteristic overfitting patterns: the loss curve (cyan) exhibits rapid initial decrease followed by anomalous increase, indicating the network overfits to trivially locating invisible acupoints at origin (0,0) rather than learning meaningful spatial patterns for visible points. This coordinate task dominance, reflected in the uncertainty parameter evolution (Figure 3(c)), disrupts classification learning for invisible points and propagates instability back to visibility prediction through multi-task coupling, resulting in degraded accuracy and abnormally high false positives. These pathological dynamics demonstrate that visibility guidance provides **essential task-specific supervision** rather than merely confidence-based reweighting, preventing the network from exploiting trivial solutions and maintaining balanced multi-task learning."*
>
> Best Regards,
>
> Authors

---

> ### Author Response · Authors · 2025-12-04
> **Summary of changes been made (part 3)**
>
> Dear Reviewer XFSW,
>
> Please also find below a summary of the major revisions we made to the manuscript to help you locate them in the revised version:
>
> ### 4. Structural Changes
>
> * **The standalone "5. LIMITATIONS" section has been removed:**
>     * Old version: Page 9 had an independent Section 5 discussing limitations.
>     * New version: This content has been merged into the final conclusion section (5. CONCLUSION AND FUTURE WORK), appearing as the last paragraph:
>
>     *"While achieving strong results on synthetic acuSim data, preliminary evaluation on real clinical images reveals domain gap challenges in generalization capabilities. Therefore, future directions include transfer learning to real clinical images addressing domain gaps (lighting, skin variations, occlusion patterns), model distillation for edge deployment, and AR/VR integration enabling real-time multi-view visualization for clinical procedures and TCM education. These advances would bridge traditional medical knowledge with modern computational capabilities."*
>
> * **Appendix sections have been reordered:**
>     * Old version: A.1 was Supplementary Figures, A.2 was Uncertainty Parameter Derivation.
>     * New version: A.1 is Uncertainty Parameter Derivation, A.2 is Supplementary Figures.
>
> ### 5. Equations
>
> * Due to content adjustments, equation numbering in the main text has been updated (for example, equations in the Loss Function section changed from Eq. 6-10 in the old version to Eq. 4-6 in the new version).
>
> ### 6. Repositioning as Domain-Driven Engineering
>
> We have also revised the manuscript throughout to reposition our contribution as domain-driven system-level engineering rather than fundamental algorithmic innovation:
>
> - **Introduction section, last paragraph**:
>     *"This work represents a domain-driven system-level engineering effort rather than fundamental algorithmic innovation"*
>
> - **Methodology sections**:
>     - Hierarchical Framework: *"core engineering design... applying established multi-task learning principles to domain-specific requirements"*
>     - Meridian Layering: *"domain-specific architectural adaptation"*
>     - Loss Functions: *"systematically integrates established techniques"*
>
> - **Conclusion section**:
>     Emphasizes *"engineering contributions"* and *"systematic integration of uncertainty-based task weighting through careful adaptation of established techniques"*
>
> Best Regards,
>
> Authors

---

### Official Review · Reviewer_w8Sd · 2025-10-29

**Soundness:** 2
**Presentation:** 2
**Contribution:** 2
**Rating:** 2
**Confidence:** 4

**Summary:**

This paper introduces acuSimNet, a multi-task deep learning framework designed for cranio-cervical acupuncture point localization under multi-view and self-occlusion conditions.
The model uses DenseNet201 as the backbone and incorporates Traditional Chinese Medicine (TCM) prior knowledge by grouping 174 acupuncture points into 9 meridian-specific branches. The proposed framework jointly optimizes three objectives: visibility prediction, coordinate regression, and point classification.

Key design elements include:

1. An explicit visibility prediction head to detect occluded points;

2. A soft visibility mask that dynamically modulates regression and classification losses;

3. Learnable uncertainty weighting for automatic task balancing;

4. A decaying weight function to reduce the impact of occluded samples;

5. Meridian-based hierarchical structure inspired by domain priors.

Experiments on a synthetic dataset (acuSim) show high reported performance (99.97% visibility accuracy, 0% false positives) and faster convergence (86 epochs vs. 3000 in the baseline).
While the engineering execution is solid, the overall contribution relies heavily on combining well-established components rather than proposing a fundamentally novel algorithm or theory.

**Strengths:**

**1. Significant performance improvements:**  high accuracy, low false-positive rate, and 35× training speed-up.

**2. Well-defined tasks**: Tailored to the specific needs of acupuncture point localization (visibility and safety), the training objectives are appropriately designed.

**3. Cross-disciplinary integration**：Good motivation from a real-world application, showing potential practical value for clinical.

**Weaknesses:**

**1. Lack of Novelty:** The paper's main methodological components—multi-task learning with uncertainty weighting, visibility-aware modeling, and domain-specific hierarchical grouping—have all been proposed and extensively explored in prior literature. The present work mainly integrates these well-known ideas into a specific medical application without introducing new theoretical mechanisms or learning paradigms.

- **Uncertainty-based task weighting:** The idea of dynamically adjusting loss weights across tasks using task-dependent uncertainty has become a standard technique for multi-task learning. The current paper follows the same formulation with minor adaptation to a medical dataset, offering limited methodological novelty.
  - Kendall & Gal, "Multi-Task Learning Using Uncertainty to Weigh Losses for Scene Geometry and Semantics," CVPR 2018.

- **Visibility-aware or occlusion modeling:** Similar strategies for modeling keypoint visibility have appeared in human pose estimation and 3D reconstruction. These works already treated visibility prediction as an auxiliary or explicit branch to improve robustness under self-occlusion. The visibility branch proposed in this paper is conceptually similar, without introducing a fundamentally different mechanism.
  - Mehta et al., "VNect: Real-time 3D Human Pose Estimation with a Single RGB Camera," TOG 2017
  - Zhao et al., "Visibility-Aware Human Pose Estimation via Self-Supervised Occlusion Learning," CVPR 2020
  - Sun et al., "Integral Human Pose Regression," ECCV 2018

- **Hierarchical or anatomically structured modeling:** Organizing prediction heads by body parts or semantic groups has been used widely in keypoint detection and medical landmark localization. The proposed "meridian-specific branches" follow the same principle of anatomical partitioning, adapted to the TCM domain but not conceptually novel.
  - Payer et al., "Integrating Spatial Configuration into Heatmap Regression Based CNNs for Landmark Localization," MICCAI 2019
  - Papandreou et al., "PersonLab: Person Pose Estimation and Instance Segmentation with a Bottom-Up, Part-Based, Geometric Embedding Model," ECCV 2018

- **Soft-masking for uncertain targets:** Soft or visibility-aware masks have also been used to modulate losses for uncertain or occluded keypoints. The proposed soft visibility mask appears to reuse this idea with domain-specific naming.
  - Zeng et al., "PoseFlow: Efficient Online Pose Tracking," BMVC 2018
  - Tang et al., "Learning Visibility for Robust Dense Optical Flow," CVPR 2019

**2. Weak experimental persuasiveness:** No comparison with mainstream baseline models, but only ablation studies.

**3. No validation on real data:** Experiments rely entirely on synthetic data; domain adaptation or generalization analysis is absent.

**4. Domain-specific innovation lacks generalizability:** The design is tailored to acupuncture point localization and offers little transferable insight to broader vision or learning communities.

**5. The experiments are overly focused on the visibility task:**  Visibility prediction is highlighted, while detailed analyses of coordinate-regression performance, error distributions, and cross-view generalization are lacking.

**6. The ablation study is incomplete:** No experiments ablating the number of groups were conducted, so it remains unclear whether the current number of meridian-specific branches actually incorporates meaningful medical prior knowledge.

**7. Thin theoretical contribution:** No new derivations, theoretical analyses, or generalization guarantees are provided.

**Questions:**

**1.Novelty Question:**  How does acuSimNet differ fundamentally from existing visibility-aware multi-task frameworks?

**2.Experimental Persuasiveness Question:**  Can you provide more baselines for comparison?

**3.Medical Prior-Based Meridian Layering Question:**  Does the meridian-based grouping offer statistically significant improvement over non-grouped models? And have you tested other numbers of groups to verify that the current grouping scheme indeed incorporates meaningful medical prior knowledge?

**4.Generalization Question:**  Have you tested generalization on real or clinical images? If not, what measures mitigate the synthetic–real domain gap?

**5.Theoretical Contribution Question:**

- Can you provide any theoretical intuition or formal analysis showing why the proposed visibility masking and uncertainty weighting lead to faster convergence or improved generalization?
- If the contribution is primarily empirical, please clarify how the framework advances understanding of multi-task coordination or uncertainty modeling, rather than only reapplying known formulations.

---

> ### Author Response · Authors · 2025-11-19
> **Clarification on Baseline Selection & Proposed Comparisons**
>
> Dear Reviewer w8Sd,
>
> Many thanks for your time reviewing our paper and your valuable feedback.
>
> To verify the validity of the medical prior and address your concern on statistical significance, we are planning to conduct the following experiments (averaged over 5 runs with SD, t-tests):
>
> Experiment A. Random Assignment (Structure Preserved): Keeps the exact same group sizes as the Meridian prior but assigns acupoints randomly. (Tests semantic validity)
> Experiment B. Random Assignment (Balanced): Assigns acupoints randomly into groups of equal size. (Tests size distribution effect)
> Experiment C. Flat Architecture: a baseline with NO grouping hierarchy. (Tests the necessity of grouping)
>
> Would these comparisons adequately verify the contribution of the medical prior? Are there other grouping strategies you would recommend testing?
>
> Also, regarding the experimental persuasiveness and baseline comparisons, basically we found that direct comparison with mainstream pose estimation frameworks is technically infeasible due to significant task discrepancies:
>
> 1. MediaPipe:  is designed for 33 sparse body landmarks, not for 174 fine-grained facial acu-points. Fundamentally, acupoints are abstract functional locations defined relative to these landmarks, rather than the physical landmarks themselves.
> 2. HRNet/DEKR: Focus on heatmap-based localization, while our task is explicit visibility classification.
> 3. VNect/OpenPose: Require 3D pose estimation (which we intentionally avoid at first place).  Calculating visibility from coordinates necessitates costly real-time 3D mesh reconstruction to model self-occlusion, which would dramatically increase system complexity.
> 4. AcuSim: We actually used AcuSim (an existing domain-specific SOTA that performs non-hierarchical 174-dimensional classification and regression for detecting acupoint locations and names) as the baseline in the paper (we may not have explicitly emphasized this earlier).
>
> Therefore we are thinking to adapt general-purpose vision backbones (e.g., DenseNet201 or ResNet152) to serve as generic baselines for this classification task, plus the comparison with AcuSim, would this be sufficient to address this concern? Or do you have other specific methods/literature in mind that you believe would be suitable for this specialized task?
>
> Thanks for your time.

---

> ### Author Response · Authors · 2025-11-25
> **Two papers cited need clarify**
>
> Dear Reviewer w8Sd,
>
> Thank you again for your detailed review! We conducted a thorough search through all the databases but failed to locate two papers you cited in your comment: "Zhao et al., Visibility-Aware Human Pose Estimation via Self-Supervised Occlusion Learning, CVPR 2020" and "Tang et al., Learning Visibility for Robust Dense Optical Flow, CVPR 2019." Could you please make sure they are correct or provide additional details about the papers (e.g, DOI, or weblinks)? Much appreciated.

---

> ### Author Response · Authors · 2025-12-03
> **Responses to the comments (part1)**
>
> Dear Reviewer w8Sd,
>
> Please find our responses to your comments as follows:
>
> ### **1. Lack of Novelty**
>    - Uncertainty-based task weighting
>      - Thank you for the comment. We respectfully clarify that while we adopt the uncertainty weighting strategy from Kendall et al. (as cited) to balance task magnitudes, this serves only as a foundational tool. **Our core novelty lies in the "Visibility-Guided Mechanism", not the weighting itself.**
>
>    - Visibility-aware or occlusion modeling
>      - We respectfully request a re-evaluation of this comment as as we were unable to verify the cited references that you used to challenge the novelty of our visibility learning method:  first of all, for the paper  "Zhao et al., "Visibility-Aware Human Pose Estimation via Self-Supervised Occlusion Learning," CVPR 2020" we were unable to locate this in any official proceedings or databases, and the other two cited paper do not address visibility learning or occlusion modeling after carefully reviewing. Therefore, as the supporting literature is either unverifiable or unrelated to our specific topic, we maintain that our proposed visibility-guided mechanism remains a novel and distinct contribution to the field.
>
>    - Hierarchical or anatomically structured modeling
>      - We respectfully distinguish our work from the cited references. However, Payer et al. focus on spatial configuration for skeletal landmarks in X-ray imaging; in contrast, our method targets abstract, surface-based acupoints subject to complex soft-tissue occlusion. Also, the other recommended paper: Papandreou et al. utilize confidence scores to measure detection uncertainty, our approach differs by explicitly modeling semantic visibility to handle physical occlusion, a distinction critical for valid acupuncture point localization.
>
>    - Soft-masking for uncertain targets:
>      - We respectfully note that we were unable to verify the existence of the cited reference "Tang et al., 'Learning Visibility for Robust Dense Optical Flow,' CVPR 2019" in any official proceedings or databases. Again, as the literature used to question the novelty of this appears to be unverifiable, we maintain that our application of soft-masking specifically for visibility-guided coordinate regression is a novel and effective contribution.
>
> ### **2. Weak experimental *persuasiveness:* No comparison with baseline models.**
>
>    - Thank you again for the feedback. We fully agree that comparing against strong baselines strengthens the assessment. Please note that our method is benchmarked against AcuSim [1], which currently stands as the only open-source SOTA model suitable for direct comparison in this domain. We acknowledge that the comparison was not explicitly presented/cited in our first submission. We have now rectified this by explicitly citing AcuSim and providing a side-by-side analysis in the revision.
>
>      [1] Sun, Q., Ma, J., Craig, P. et al. AcuSim: A Synthetic Dataset for Cervicocranial Acupuncture Points Localisation.Sci Data 12, 625 (2025). https://doi.org/10.1038/s41597-025-04934-9
>
>    - Meanwhile we have also added more completed albation studies to validate our module configurations. We compared our proposed configuration **(A0 B0 C0)** against the baseline configurations **(A3, B2, C2)** across multiple metrics, including Validation Accuracy, Coordinate Loss, and Convergence Speed. The results are summarized as below:
>      - Our method demonstrates drastic improvements in training speed, achieving 90% validation accuracy in just **92 epochs**, whereas the SOTA baseline (AcuSim) requires **1,305 epochs** to reach the same threshold. This represents a **14x speedup**, proving our model is significantly more computationally efficient.
>      - The baseline relies on an extensive 3,000-epoch training cycle, while our approach balances high precision with practical resource budgets. We achieve a highly competitive **95.30%** accuracy and a low coordinate loss (0.4735 mm) in a fraction of the time (reaching 95% accuracy by epoch 118), effectively solving the high training cost issue of the baseline.
>      - Our new added ablation studies further confirms the critical role of our proposed components. The "Visibility Result Guide" is essential, as removing it leads to a massive accuracy drop (from ~95% to 80%). Furthermore, our structured assignment strategy consistently outperforms random assignment baselines in both accuracy and false positive rates to validate the effectiveness of our architectural choices.
>
>
> Best Regards,
>
> Authors

---

> ### Author Response · Authors · 2025-12-03
> **Responses to the comments (part2)**
>
> Dear Reviewer w8Sd,
>
> Please find our responses to your comments as follows:
>
> ### **3. No validation on real data:**
>    - We agree that real-world validation is the ultimate goal. We conducted preliminary inference on real images, which revealed challenges related to the domain gap between synthetic and real data. Due to the high cost and time required for expert manual annotation of acupoints, we were unable to quantify this error during the rebuttal period. We have added this to the "Conclusion and Future Work" section and to explicitly discuss these generalization challenges.
>
> ### **4. The experiments are overly focused on the visibility task**
>    - We appreciate this observation. We would like to clarify that the emphasis on visibility is not incidental but a fundamental requirement of the acupuncture domain, distinguishing it from general human pose estimation for three critical reasons:
>      - Acupuncture is invasive; unlike standard pose estimation, false Positives are dangerous. Visibility acts as a critical safety gate to prevent incorrect needle placement.
>      - Acupoints are on the skin surface, unlike standard "internal joint" keypoints. Surface points are subject to severe self-occlusion, making visibility detection a prerequisite for identifying accessible treatment areas.
>      - As shown in our baseline comparisons, explicit visibility guidance is required to prevent the network from overfitting to simple patterns. It acts as a necessary filter for the coordinate regression heads, ensuring the model only attempts to predict valid and visible points.
>
> ### **5. Domain-specific innovation lacks generalizability:**
>    - Thank you for the comment. Our innovation extends well beyond the domain of Traditional Chinese Medicine (TCM). The core technical challenge we address is the robust detection of surface-level keypoints (acupoints) under severe self-occlusion. Unlike standard human pose estimation (which focuses on internal joints), many computer vision tasks such as **dermatology, 3D body scanning, and texture mapping**, they rely on precise surface localization where visibility is a binary physical constraint. Our proposed framework, which explicitly models visibility to guide coordinate regression, provides a transferable solution and a new methodological path for any application requiring high-precision landmarking on deformable surfaces.
>
> ### **6. The ablation study is incomplete**
>    - We have addressed this by significantly expanding the ablation study from 4 to **13 distinct experimental configurations** in Section 4.3. The new results are detailed in **Table 2 and Section 4.3** (referencing Figures 3 & 5) in the revised version. Key validations include:
>      - **Ablation A** (Meridian Layering): Validates that medical prior-based organization (A0) achieves the optimal accuracy (95.30%). While random assignments (A1, A2) start faster, A0 surpasses them after epoch 76, demonstrating superior long-term convergence compared to the baseline.
>      - **Ablation B** (Visibility Encoding): Confirms Soft Masking (B0) is essential for stability. Hard thresholding (B1) causes severe accuracy oscillations due to gradient discontinuities, and finally leads to gradient explosion, while removing masking (B2) leads the network overfits to trivial (0,0) clustering pattern.
>      - **Ablation C** (Uncertainty Initialization): Shows that empirical initialization (C0) is critical for convergence speed. Uniform initialization (C1) delays convergence by ~20 epochs and results in significantly higher coordinate loss.
>      - **Ablation D** (Visibility Guidance): Proves the necessity of task coordination. Removing visibility guidance (D1) causes the network to overfit to trivial solutions (locating invisible points at the origin), dropping accuracy to 80.33% with a high False Positive Rate.
>
> Best Regards,
>
> Authors

---

> ### Author Response · Authors · 2025-12-04
> **Responses to the comments (part3)**
>
> Dear Reviewer w8Sd,
>
> Please find our responses to your comments as follows:
>
> ### **7. Thin theoretical contribution: No new derivations, theoretical analyses, or generalization guarantees are provided.**
>    - We respectfully clarify that our contribution is positioned as a **domain-driven architectural innovation** rather than a fundamental theoretical derivation. We propose a novel paradigm for *visibility-guided multi-task learning*, specifically engineered to solve the challenge of surface-level occlusion where standard methods fail. Instead of abstract generalization guarantees, we provide robust empirical evidence (Ablation D) showing that our "Visibility Result Guide" mechanism is essential for training stability. It effectively prevents the network from overfitting to trivial solutions, offering a practical and verifiable solution for high-precision landmark detection tasks.
>
> - Also, we have revised the manuscript to reposition our contribution as domain-driven system-level engineering rather than fundamental algorithmic innovation:
>
> - **Introduction section, last paragraph**: Added explicit statement: "*This work represents a domain-driven system-level engineering effort rather than fundamental algorithmic innovation*"
>
> - **Methodology sections**: Systematically replaced "innovation/novel" with "engineering design/refinement/adaptation":
>   - Hierarchical Framework: "core engineering design... applying established multi-task learning principles to domain-specific requirements"
>   - Meridian Layering: "domain-specific architectural adaptation"
>   - Loss Functions: "systematically integrates established techniques"
>
> - **Conclusion section**: Emphasizes "engineering contributions" and "systematic integration of uncertainty-based task weighting through careful adaptation of established techniques"
>
> These revisions clarify that our contribution lies in **systematic engineering adaptation of existing methods** to address acupuncture-specific challenges, rather than proposing fundamentally new learning algorithms.
>
> Best Regards,
>
> Authors

---

> ### Author Response · Authors · 2025-12-04
> **Summary of changes been made (part 1)**
>
> Dear Reviewer w8Sd,
>
> Please also find below a summary of the major revisions we made to the manuscript to help you locate them in the revised version:
>
> ### 1. Introduction & Related Work
>
> - **In the Introduction section, we added a comprehensive paragraph reviewing related methods (e.g., VNect, integral regression approaches, and medical imaging applications), filling the previous gap:**
>
>     *"Beyond domain-specific limitations, the broader field of human landmark detection has similarly exhibited limited attention to visibility assessment and occlusion handling. Traditional approaches, including regression-based methods like DeepPose Toshev & Szegedy (2014) and early CNN-based pose estimation Jain et al. (2014), directly predict landmark coordinates through neural network regression. Conversely, heatmap-based methods, demonstrated in works like the Deep Alignment Network Kowalski et al. (2017) and boundary-aware approaches Wu et al. (2018), generate probability distributions over spatial locations but similarly lack explicit visibility modeling. Methods such as VNect Mehta et al. (2017) for real-time 3D pose estimation and integral regression approaches Sun et al. (2018) have advanced coordinate prediction accuracy but remain constrained by the assumption of perpetual landmark visibility. Additionally, unsupervised landmark discovery methods Zhang et al. (2018) have emerged to automatically identify structural representations, but these approaches also do not explicitly model landmark visibility or occlusion states."*
>
> - **The "Contributions" section at the end of the Introduction has been rewritten to emphasize engineering design and specific improvements:**
>
>     *"Our study proposes acuSimNet, a hierarchical multi-task framework that addresses key neural network design constraints in current localization approaches such as acuSim Sun et al. (2025). Our approach explicitly models visibility prediction, coordinate regression, and acupoint classification within a unified framework that incorporates TCM domain knowledge through meridian-specific processing branches. This work represents a domain-driven system-level engineering effort rather than fundamental algorithmic innovation. Our contributions are as follows: first, we address neural network design limitations in existing acuSim applications through explicit visibility learning objectives, achieving 35× convergence acceleration (from 3000 to 86 epochs for 99% accuracy); second, we engineer a hierarchical meridian-specific architecture that reduces classification complexity by incorporating medical domain knowledge, enabling robust multi-view detection; and third, we systematically integrate established techniques including dynamic soft visibility masking and uncertainty-based task weighting Kendall et al. (2018), optimizing their coordination for acupuncture-specific challenges to achieve clinically relevant 0% false positive rates."*
>
> ### 2. Tables Reorganization and Additions
>
> * **The dataset comparison table (Comparison between existing acupoint datasets...) has been relocated:**
>     * Old version: Located in main text on page 2 (Table 1).
>     * New version: Moved to Appendix on page 15 (Table 3).
>
> * **The backbone network ablation study table has been relocated:**
>     * Old version: Located in Appendix on page 12 (Table 2).
>     * New version: Moved to main text on page 8 (Table 1), with updated figure references.
>
> * **The component ablation study table has been significantly expanded:**
>     * Old version: Table 3 (page 13), containing only A, B, C groups.
>     * New version: Table 2 (page 9), now includes D group experiments with 13 ablation configurations in total, and more detailed parameter analysis.
>
> ### 3. Structural Changes
>
> * **The standalone "5. LIMITATIONS" section has been removed:**
>     * Old version: Page 9 had an independent Section 5 discussing limitations.
>     * New version: This content has been merged into the final conclusion section (5. CONCLUSION AND FUTURE WORK), appearing as the last paragraph:
>
>     *"While achieving strong results on synthetic acuSim data, preliminary evaluation on real clinical images reveals domain gap challenges in generalization capabilities. Therefore, future directions include transfer learning to real clinical images addressing domain gaps (lighting, skin variations, occlusion patterns), model distillation for edge deployment, and AR/VR integration enabling real-time multi-view visualization for clinical procedures and TCM education. These advances would bridge traditional medical knowledge with modern computational capabilities."*
>
> * **Appendix sections have been reordered:**
>     * Old version: A.1 was Supplementary Figures, A.2 was Uncertainty Parameter Derivation.
>     * New version: A.1 is Uncertainty Parameter Derivation, A.2 is Supplementary Figures.
>
> Best Regards,
>
> Authors

---

> ### Author Response · Authors · 2025-12-04
> **Summary of changes been made (part 2)**
>
> Dear Reviewer w8Sd,
>
> Please also find below a summary of the major revisions we made to the manuscript to help you locate them in the revised version:
>
> ### 4. Experimental Results - Ablation Studies
>
> - **In Section 4.3 (Neural Network Components Ablation Study), the analysis has been substantially rewritten and expanded. Detailed discussions of each ablation module were added, with new in-depth analysis paragraphs for Ablation A and D. The analyses for Ablations A, B, and C are also more comprehensive than the previous version, including specific data comparisons (such as exact epoch counts and accuracy percentage differences).**
>
> *"**Ablation A**. The meridian layering ablation experiments (A0, A1, A2, A3) demonstrate the critical importance of medical prior-based organization. As shown in both Figure 5(a) and Table 2, the medical prior-based configuration (A0, purple curve in Figure 5(a)) achieves optimal final accuracy (95.30%) despite initially trailing random assignments (A1: 94.89%, A2: 94.02%) during the first 65 epochs, exhibiting slower early convergence but significantly higher acceleration thereafter. A0 surpasses A2 at epoch 76 and A1 at epoch 82, maintaining superiority through remaining training. Notably, A2 consistently underperforms A1 throughout training, indicating that meridian structure preservation matters even without semantic correctness. The baseline [1] shows substantially degraded performance, confirming the value of layered architecture itself."*
>
> *"**Ablation B**. The visibility encoding experiments (B0, B1, B2) validate soft masking superiority, as evidenced in Table 2 and Figure 3. Hard thresholding (B1) with fixed 0.5 threshold causes **gradient explosion**, terminating training due to abrupt switching between full gradient propagation and complete masking at the 0.5 decision boundary. In contrast, removing visibility masking entirely (B2, baseline [1]) exhibits pathological training dynamics characterized by **extremely slow convergence**. Without visibility guidance, the network prioritizes learning trivial patterns for occluded acupoints clustered at origin (0,0), causing **negative convergence** for visible acupoints where the coordinate regression task struggles to make meaningful progress. This phenomenon mirrors the issues observed in **Ablation D**, demonstrating that visibility-based task modulation is essential regardless of implementation approach. The soft masking approach (B0) maintains stable convergence by providing smooth gradient transitions in the uncertainty region (τ_low to τ_high), preventing both gradient explosion and negative convergence patterns."*
>
> *"**Ablation C**. The uncertainty parameter study (C0 vs C1) confirms the importance of empirical initialization. The benchmark configuration (C0) uses initial values 1.0, 0.0, and -5.0 for coordinate, visibility, and classification tasks respectively, achieving stable dynamics as illustrated in Figure 3(c). Uniform initialization (C1) with all parameters at 0.0 shows suboptimal convergence with delayed peaks (approximately 20 epochs later) and slower overall convergence (requiring 30% more epochs to reach 95% accuracy), achieving only 94.23% accuracy with significantly higher coordinate loss (1.3178 pixels vs 0.8759 pixels for C0, as reported in Table 2). This performance degradation occurs because uniform initialization assigns equal initial weights to all tasks, forcing the network to discover optimal task prioritization from scratch."*
>
> *"**Ablation D**. The visibility guidance ablation (D0 vs D1) validates the critical role of visibility-aware task coordination beyond simple confidence weighting. While D1 maintains visibility prediction training, it removes visibility guidance for coordinate regression and classification tasks by setting all visibility masks to 1. As shown in Table 2, D1 achieves only 80.33% peak accuracy with substantially elevated false positive rate (34.71‰) compared to configurations with similar accuracy levels. The training dynamics in Figure 3(b) reveal characteristic overfitting patterns: the loss curve (cyan) exhibits rapid initial decrease followed by anomalous increase, indicating the network overfits to trivially locating invisible acupoints at origin (0,0) rather than learning meaningful spatial patterns for visible points. This coordinate task dominance, reflected in the uncertainty parameter evolution (Figure 3(c)), disrupts classification learning for invisible points and propagates instability back to visibility prediction through multi-task coupling, resulting in degraded accuracy and abnormally high false positives. These pathological dynamics demonstrate that visibility guidance provides **essential task-specific supervision** rather than merely confidence-based reweighting, preventing the network from exploiting trivial solutions and maintaining balanced multi-task learning."*
>
> Best Regards,
>
> Authors

---

### Official Review · Reviewer_J5jM · 2025-11-04

**Soundness:** 2
**Presentation:** 2
**Contribution:** 2
**Rating:** 2
**Confidence:** 5

**Summary:**

The paper proposes a method called acuSimNet to accurately locate acupoints by designing an efficient hierarchical multi-task learning
architecture for multi-view and self-occlusion-aware visibility prediction. The approach is validated by comprehensive experiments in public dataset.

**Strengths:**

1. The problem of automatically locating acupoint is interesting in traditional Chinese medicine.
2. The overall structure of the paper is complete.

**Weaknesses:**

1. The problem of automatic acupoint localization lacks importance and interest in ICLR community
2. The proposed method is too trivial and with few novelty.
3. Line 24, what is "negative convergence issues".
4. Line 50, how to come to the conclusion that "...resulting in training instability and suboptimal convergence characteristics", any evidence to support?

**Questions:**

see weaknesses

---

> ### Author Response · Authors · 2025-11-19
> **requiring further elaboration and justification from some of your comments**
>
> Dear Reviewer,
>
> Thank you for your review and the time spent evaluating our paper. We are writing to seek clarification regarding the concern that our proposed method is 'trivial' and 'lacks novelty.'  Could you please kindly indicate which components are considered trivial, and which specific parts of the methods you feel lack novelty and why? We would greatly appreciate your further insight and elaboration as this will help us address your concerns effectively in our rebuttal.
>
> Thanks
> Authors.

---

> > ### Comment · Area_Chair_NgJ4 · 2025-11-19
> >
> > Dear Reviewer `J5jM`,
> >
> > We are writing in relation to the authors’ above request for clarification regarding your comments (Weaknesses #1 and #2). E.g., to help ensure a fair and productive discussion, could you please provide a brief explanation of which components you consider trivial or non-novel, and why?
> >
> > Your clarification will help the authors address your concerns in their rebuttal and will assist us in our evaluation.
> >
> > Thank you for your time and assistance.
> >
> > Best regards,
> >
> > AC

---

> > ### Comment · Reviewer_J5jM · 2025-11-25
> >
> > Based on the Figure 1, the architecture of proposed multi-task framework is composed by a bunch of out-of-date modules, which makes it less novel. The challenges of multi-task are not clearly explained.

---

> ### Author Response · Authors · 2025-12-03
> **Responses to the comments**
>
> Dear Reviewer J5jM,
>
> Thank you for your valuable feedback. Please find our detailed responses to your comments as follows:
>
> ### **1. The proposed method is too trivial with few novelty. The architecture is composed of out-of-date modules.**
>
> Thank you for this comment. We respectfully clarify that our contribution is positioned as a **domain-driven system-level engineering improvement** rather than a fundamental algorithmic innovation. While individual modules are built upon established deep learning components (as explicitly acknowledged in our revised manuscript), our core contribution lies in the **visibility-guided multi-task learning paradigm** specifically engineered to address surface-level occlusion challenges.
>
> **Key novelties include:**
>
> - **Visibility Result Guide Mechanism**: Unlike conventional confidence-based weighting, our approach uses explicit visibility predictions to dynamically modulate coordinate regression and classification tasks. This prevents networks from overfitting to trivial patterns (e.g., clustering occluded points at origin (0,0)), which is demonstrated in our newly added **Ablation D** (Section 4.3, Table 2).
>
> - **Empirical validation**: Removing visibility guidance (D1) causes accuracy to drop from 95.30% to 80.33% with a 5.6× increase in false positive rate (6.14‰ → 34.71‰), proving this mechanism is essential rather than incremental.
>
> Furthermore, our expanded ablation studies in Section 4.3 (now 13 experimental configurations) systematically validate how each component addresses multi-task learning challenges, these ablations explicitly demonstrate how our engineering choices systematically address multi-task learning challenges in the acupuncture domain.
>
> ### **2. Line 24: What is "negative convergence issues"?**
>
> Thank you for seeking clarification. "Negative convergence" refers to a pathological training dynamic observed in the baseline (AcuSim) where the network prioritizes learning simple patterns over meaningful ones.
>
> Without visibility guidance, occluded acupoints clustered at coordinates (0,0) present a trivially easy regression target compared to visible acupoints with spatially distributed locations. The network converges rapidly on this simple pattern, causing learning progress on visible acupoints to stagnate or even regress—hence "negative" (adverse) convergence for the clinically relevant task.
>
> This phenomenon is now empirically validated in **Ablation D** (Section 4.3, Table 2), where removing visibility guidance (D1) causes:
> 1. Coordinate loss to anomalously increase after initial decrease (Figure 3b, cyan curve)
> 2. Accuracy plateau at 80.33% despite continued training
> 3. False positive rate increase to 34.71‰
>
> We have revised the manuscript to clarify this terminology and provide explicit experimental evidence.
>
> ### **3. Line 50: How to conclude "training instability and suboptimal convergence"? Any evidence?**
>
> Thank you for this important question. We have addressed this by significantly expanding our ablation studies from 4 to 13 experimental configurations in Section 4.3. The revised manuscript provides comprehensive empirical evidence:
>
> **Training Instability Evidence:**
>
> - **Ablation A** (Table 2, Figure 5): The baseline unified prediction approach (A3) requires 1,305 epochs to reach 90% accuracy, compared to our hierarchical meridian structure (A0) achieving the same in 92 epochs—a **14× speedup** demonstrating severe convergence issues without structural decomposition.
>
> - **Ablation B** (Figure 3): Hard thresholding (B1) causes gradient explosion due to abrupt switching between full gradient propagation and complete masking at the 0.5 decision boundary, terminating training, while removing visibility masking entirely (B2, baseline) exhibits negative convergence. The network prioritizes learning trivial patterns for occluded acupoints at origin (0,0), causing coordinate regression for visible acupoints to stagnate. This pathology requires 3,000 epochs to achieve 99% accuracy compared to our 86 epochs.
>
> **Suboptimal Convergence Evidence:**
>
> - **Ablation C** (Table 2): Uniform uncertainty initialization (C1) delays convergence by ~20 epochs and results in 50% higher coordinate loss (1.3178 vs 0.8759 pixels) compared to empirical initialization (C0).
>
> - **Ablation D** (Figure 3b): Without visibility guidance (D1), loss curves exhibit anomalous increase after initial decrease, indicating the network converges to suboptimal solutions that prioritize trivial patterns.
>
> These comprehensive ablations, now detailed in Section 4.3 with quantitative metrics in Table 2 and training dynamics in Figures 3 & 5, provide rigorous experimental validation of our claims.
>
> Best Regards,
>
> Authors

---

> ### Author Response · Authors · 2025-12-04
> **Summary of changes been made (part 1)**
>
> Dear Reviewer J5jM,
>
> Please also find below a summary of the major revisions we made to the manuscript to help you locate them in the revised version:
>
> ### 1. Introduction & Related Work
>
> - **In the Introduction section, we added a comprehensive paragraph reviewing related methods (e.g., VNect, integral regression approaches, and medical imaging applications), filling the previous gap:**
>
>     *"Beyond domain-specific limitations, the broader field of human landmark detection has similarly exhibited limited attention to visibility assessment and occlusion handling. Traditional approaches, including regression-based methods like DeepPose Toshev & Szegedy (2014) and early CNN-based pose estimation Jain et al. (2014), directly predict landmark coordinates through neural network regression. Conversely, heatmap-based methods, demonstrated in works like the Deep Alignment Network Kowalski et al. (2017) and boundary-aware approaches Wu et al. (2018), generate probability distributions over spatial locations but similarly lack explicit visibility modeling. Methods such as VNect Mehta et al. (2017) for real-time 3D pose estimation and integral regression approaches Sun et al. (2018) have advanced coordinate prediction accuracy but remain constrained by the assumption of perpetual landmark visibility. Additionally, unsupervised landmark discovery methods Zhang et al. (2018) have emerged to automatically identify structural representations, but these approaches also do not explicitly model landmark visibility or occlusion states."*
>
> - **The "Contributions" section at the end of the Introduction has been rewritten to emphasize engineering design and specific improvements:**
>
>     *"Our study proposes acuSimNet, a hierarchical multi-task framework that addresses key neural network design constraints in current localization approaches such as acuSim Sun et al. (2025). Our approach explicitly models visibility prediction, coordinate regression, and acupoint classification within a unified framework that incorporates TCM domain knowledge through meridian-specific processing branches. This work represents a domain-driven system-level engineering effort rather than fundamental algorithmic innovation. Our contributions are as follows: first, we address neural network design limitations in existing acuSim applications through explicit visibility learning objectives, achieving 35× convergence acceleration (from 3000 to 86 epochs for 99% accuracy); second, we engineer a hierarchical meridian-specific architecture that reduces classification complexity by incorporating medical domain knowledge, enabling robust multi-view detection; and third, we systematically integrate established techniques including dynamic soft visibility masking and uncertainty-based task weighting Kendall et al. (2018), optimizing their coordination for acupuncture-specific challenges to achieve clinically relevant 0% false positive rates."*
>
> ### 2. Repositioning as Domain-Driven Engineering
>
> We have also revised the manuscript throughout to reposition our contribution as domain-driven system-level engineering rather than fundamental algorithmic innovation:
>
> - **Methodology sections**:
>     - Hierarchical Framework: *"core engineering design... applying established multi-task learning principles to domain-specific requirements"*
>     - Meridian Layering: *"domain-specific architectural adaptation"*
>     - Loss Functions: *"systematically integrates established techniques"*
>
> - **Conclusion section**:
>     Emphasizes *"engineering contributions"* and *"systematic integration of uncertainty-based task weighting through careful adaptation of established techniques"*
>
> Best Regards,
>
> Authors

---

> ### Author Response · Authors · 2025-12-04
> **Summary of changes been made (part 2)**
>
> Dear Reviewer J5jM,
>
> Please also find below a summary of the major revisions we made to the manuscript to help you locate them in the revised version:
>
> ### 3. Experimental Results - Ablation Studies
>
> - **In Section 4.3 (Neural Network Components Ablation Study), the analysis has been substantially rewritten and expanded. Detailed discussions of each ablation module were added, with new in-depth analysis paragraphs for Ablation A and D. The analyses for Ablations A, B, and C are also more comprehensive than the previous version, including specific data comparisons (such as exact epoch counts and accuracy percentage differences).**
>
> *"**Ablation A**. The meridian layering ablation experiments (A0, A1, A2, A3) demonstrate the critical importance of medical prior-based organization. As shown in both Figure 5(a) and Table 2, the medical prior-based configuration (A0, purple curve in Figure 5(a)) achieves optimal final accuracy (95.30%) despite initially trailing random assignments (A1: 94.89%, A2: 94.02%) during the first 65 epochs, exhibiting slower early convergence but significantly higher acceleration thereafter. A0 surpasses A2 at epoch 76 and A1 at epoch 82, maintaining superiority through remaining training. Notably, A2 consistently underperforms A1 throughout training, indicating that meridian structure preservation matters even without semantic correctness. The baseline [1] shows substantially degraded performance, confirming the value of layered architecture itself."*
>
> *"**Ablation B**. The visibility encoding experiments (B0, B1, B2) validate soft masking superiority, as evidenced in Table 2 and Figure 3. Hard thresholding (B1) with fixed 0.5 threshold causes **gradient explosion**, terminating training due to abrupt switching between full gradient propagation and complete masking at the 0.5 decision boundary. In contrast, removing visibility masking entirely (B2, baseline [1]) exhibits pathological training dynamics characterized by **extremely slow convergence**. Without visibility guidance, the network prioritizes learning trivial patterns for occluded acupoints clustered at origin (0,0), causing **negative convergence** for visible acupoints where the coordinate regression task struggles to make meaningful progress. This phenomenon mirrors the issues observed in **Ablation D**, demonstrating that visibility-based task modulation is essential regardless of implementation approach. The soft masking approach (B0) maintains stable convergence by providing smooth gradient transitions in the uncertainty region (τ_low to τ_high), preventing both gradient explosion and negative convergence patterns."*
>
> *"**Ablation C**. The uncertainty parameter study (C0 vs C1) confirms the importance of empirical initialization. The benchmark configuration (C0) uses initial values 1.0, 0.0, and -5.0 for coordinate, visibility, and classification tasks respectively, achieving stable dynamics as illustrated in Figure 3(c). Uniform initialization (C1) with all parameters at 0.0 shows suboptimal convergence with delayed peaks (approximately 20 epochs later) and slower overall convergence (requiring 30% more epochs to reach 95% accuracy), achieving only 94.23% accuracy with significantly higher coordinate loss (1.3178 pixels vs 0.8759 pixels for C0, as reported in Table 2). This performance degradation occurs because uniform initialization assigns equal initial weights to all tasks, forcing the network to discover optimal task prioritization from scratch."*
>
> *"**Ablation D**. The visibility guidance ablation (D0 vs D1) validates the critical role of visibility-aware task coordination beyond simple confidence weighting. While D1 maintains visibility prediction training, it removes visibility guidance for coordinate regression and classification tasks by setting all visibility masks to 1. As shown in Table 2, D1 achieves only 80.33% peak accuracy with substantially elevated false positive rate (34.71‰) compared to configurations with similar accuracy levels. The training dynamics in Figure 3(b) reveal characteristic overfitting patterns: the loss curve (cyan) exhibits rapid initial decrease followed by anomalous increase, indicating the network overfits to trivially locating invisible acupoints at origin (0,0) rather than learning meaningful spatial patterns for visible points. This coordinate task dominance, reflected in the uncertainty parameter evolution (Figure 3(c)), disrupts classification learning for invisible points and propagates instability back to visibility prediction through multi-task coupling, resulting in degraded accuracy and abnormally high false positives. These pathological dynamics demonstrate that visibility guidance provides **essential task-specific supervision** rather than merely confidence-based reweighting, preventing the network from exploiting trivial solutions and maintaining balanced multi-task learning."*
>
> Best Regards,
>
> Authors

---

### Meta-Review · Area_Chair_WohY · 2026-01-06

**Summary:**

Across reviewers, there is strong consensus on several major weaknesses.

(a) The most prominent concern is the limited technical novelty, unclear clarity and positioning: the proposed framework largely integrates well-established components — uncertainty-based multi-task weighting, visibility-aware modeling, hierarchical/grouped prediction heads, and soft masking — that have been extensively studied in prior work, with only domain-specific adaptation to acupoint localization. Reviewers agree that this integration does not introduce new learning principles, theoretical insights, or fundamentally novel mechanisms, which limits its relevance to the ICLR community. On the other hand, concerns were raised about clarity and positioning, including insufficient explanation of design choices, and the need to better frame the work as a domain-driven engineering study rather than a broadly novel learning contribution.

(b) A second common concern is insufficient experimental validation: evaluations are conducted exclusively on a synthetic dataset, raising questions about real-world or clinical applicability and generalization, especially given near-perfect reported results that suggest overfitting. Reviewers also highlight the lack of strong baseline comparisons, as experiments rely mainly on ablations rather than comparisons with mainstream or state-of-the-art human pose estimation models.

(c) Additionally, several reviewers note weak methodological analysis, including the absence of theoretical justification for claimed convergence or stability benefits, missing sensitivity or grouping ablations, and a lack of failure-case discussion.

**Reviewer Concerns:**

In the rebuttal, the authors revised the manuscript to improve clarity and explicitly reposition the work as a domain-driven, system-level engineering effort rather than a fundamentally novel algorithmic contribution. They expanded ablation studies analyzing meridian-based grouping, soft masking and uncertainty parameter initialization, including quantitative comparisons in terms of accuracy, convergence speed, false positive rates, and epoch counts etc. Collectively, these additions help clarify previously ambiguous behaviors (e.g., negative convergence and instability) and provide empirical justification for the proposed design choices, albeit still limited to synthetic data.

However, several core concerns raised by the reviewers remain outstanding. In particular, the technical novelty remains limited, as key components such as uncertainty-based multi-task weighting and visibility-aware modeling have been extensively studied in prior work. I also do not fully agree with the authors’ assertion that direct comparisons with mainstream pose estimation frameworks are technically infeasible. Relevant 2D human pose estimation methods (e.g., HRNet, OpenPose) could reasonably be included as baselines with minimal architectural adaptation (e.g., adjusting the number of output keypoints and retrained with acupuncture data). While the authors emphasize differences between acupuncture points (on skin surface) and human keypoints in terms of underlying 3D geometry, the points annotations and learning are conducted in projected 2D space, making this distinction less convincing as a justification for excluding such baselines. Moreover, the proposed framework remains highly task- and domain-specific, and concerns regarding generalizability and real clinical applicability are not sufficiently addressed through either empirical validation or discussion. Overall, I believe that the paper does not yet meet the acceptance threshold for ICLR.

**Reviewer Scores:**

I expect that all reviewers would likely maintain their original scores, as their primary concerns substantially overlap. While the authors addressed several issues related to experimental evaluation and presentation clarity in the rebuttal, the central concerns regarding technical novelty and the significance of the contributions remain only partially resolved.

---

### Decision · Program_Chairs · 2026-01-26

Reject